# FLUID: SCALING AUTOREGRESSIVE TEXT-TO-IMAGE GENERATIVE MODELS WITH CONTINUOUS TOKENS

**Lijie Fan[1,*]**   **Tianhong Li[2,†]**   **Siyang Qin[1,†]**   **Yuanzhen Li[1]**   **Chen Sun[1]**
**Michael Rubinstein[1]**   **Deqing Sun[1]**   **Kaiming He[2]**   **Yonglong Tian[1,*]**
[1]Google DeepMind   [2]MIT   [*] equal contribution, project lead   [†] equal contribution

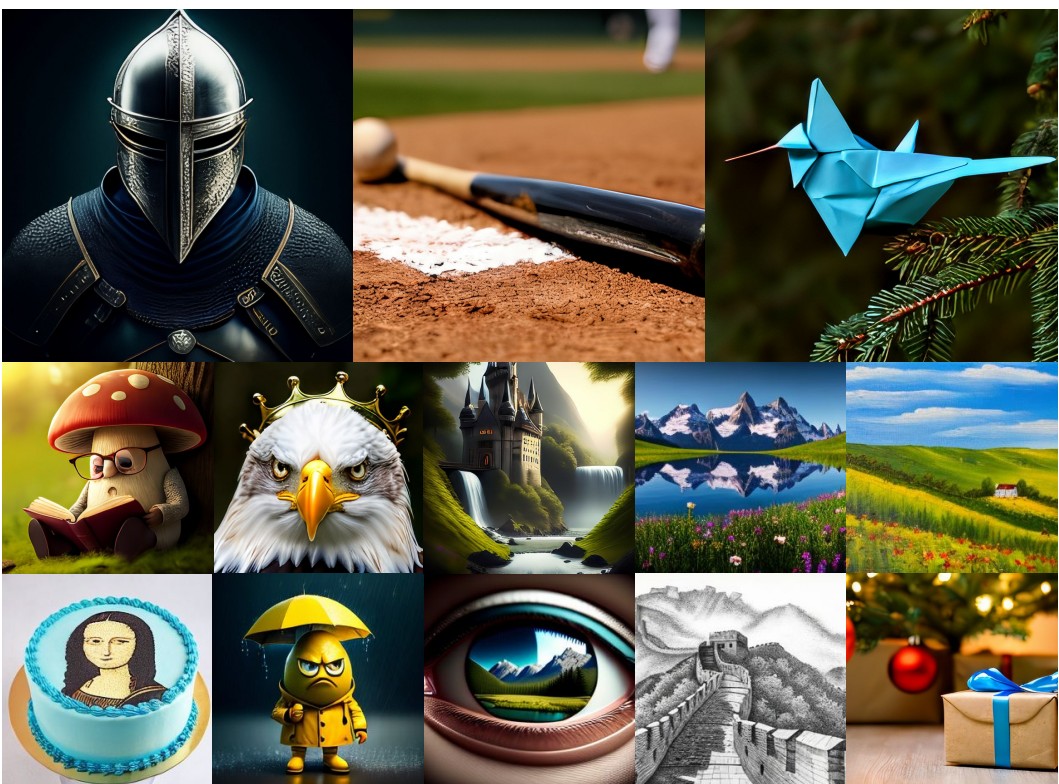

Figure 1: Samples from our Fluid 10.5B autoregressive model with continuous tokens.

## ABSTRACT

Scaling up autoregressive models in vision has not proven as beneficial as in large language models. In this work, we investigate this scaling problem in the context of text-to-image generation, focusing on two critical factors: whether models use discrete or continuous tokens, and whether tokens are generated in a random or fixed raster order using BERT- or GPT-like transformer architectures. Our empirical results show that, while all models scale effectively in terms of validation loss, their evaluation performance—measured by FID, GenEval score, and visual quality—follows different trends. Models based on continuous tokens achieve significantly better visual quality than those using discrete tokens. Furthermore, the generation order and attention mechanisms significantly affect the GenEval score: random-order models achieve notably better GenEval scores compared to raster-order models. Inspired by these findings, we train Fluid, a random-order autoregressive model on continuous tokens. Fluid 10.5B model achieves a new state-of-the-art zero-shot FID of 6.16 on MS-COCO 30K, and 0.69 overall score on the GenEval benchmark. We hope our findings and results will encourage future efforts to further bridge the scaling gap between vision and language models.

## 1 INTRODUCTION

Scaling laws underpin the unprecedented success of large language models (LLMs). Empirically, increasing the number of parameters in autoregressive models consistently leads to significant performance improvements and the emergence of new capabilities in natural language processing (NLP) tasks (Devlin et al., 2018; Radford et al., 2018; Brown et al., 2020; Kaplan et al., 2020; Wei et al., 2022). This empirical relationship has inspired numerous efforts to scale up language models, resulting in the development of many highly capable models (Bubeck et al., 2023; Team et al., 2023; Achiam et al., 2023).

Encouraged by this success, many attempts have been made to adopt and scale up autoregressive models in computer vision, particularly for generative tasks like text-to-image generation (Yu et al., 2021; 2023; Bai et al., 2024; El-Nouby et al., 2024). However, the performance and visual quality of content generated by these models often fall short compared to other generative models, such as diffusion models (Ho et al., 2020; Saharia et al., 2022; Rombach et al., 2022a; Esser et al., 2024), leaving it unclear whether similar scaling laws apply to the vision domain.

We propose several hypotheses for the performance gap. First, the vector quantization (VQ) (van den Oord et al., 2017) step, which is required for most visual autoregressive models, may introduce significant information loss, ultimately limiting model performance. Second, unlike the inherently sequential nature of language, generating visual content might benefit more from a different autoregressive prediction order. Third, there is often a confusion between two levels of generalizability when evaluating scaling laws in vision models: (a) generalization to new data using the same metric as the training loss (commonly referred to as validation loss), and (b) generalization to a new metric or problem different from the training objective, such as FID (Heusel et al., 2017), the GenEval benchmark (Ghosh et al., 2024), or visual quality. We hypothesize that power-law scaling (Kaplan et al., 2020) applies to (a) for autoregressive models on vision data, but not necessarily to (b).

To investigate these hypotheses, we conduct a comprehensive empirical study on the scaling behavior of autoregressive models in the context of text-to-image generation. Specifically, we explore two key factors: whether the model operates on continuous or discrete tokens, and whether tokens are generated in a random or fixed raster order. To this end, we utilize the Diffusion Loss (Li et al., 2024) to make autoregressive models compatible with continuous tokens. We generalize BERT-like vision model MaskGIT (Chang et al., 2022) as random-order autoregression, as it conceptually predicts output tokens in a randomized order while retaining the autoregressive nature of "predicting next tokens based on known ones". We analyze the behavior of four autoregressive variants, each employing different combinations of these two factors. We scale their parameters from 150M to 3B and evaluate their performance using three metrics: validation loss, FID (Heusel et al., 2017), and GenEval score (Ghosh et al., 2024). We also inspect the visual quality of the generated images.

Our experiments indicate that VQ-based models, regardless of whether they use a random or fixed raster order, exhibit a *slower* improvement in FID scores when scaling up model size compared to models operating on continuous tokens. VQ models also produce images of lower visual quality, likely due to information loss introduced by vector quantization.

Furthermore, the token generation order and the associated attention mechanism primarily influence the global structure of the generated image. In the GenEval benchmark, random-order models with bidirectional attention significantly outperform raster-order models with causal attention, particularly when generating multiple objects. Random-order models can readjust the global structure at every prediction step, whereas raster-order models cannot. This suggests that the token generation order plays a crucial role in achieving better text-to-image alignment.

Our experiments also demonstrate that validation loss scales as a power-law with model size, no matter whether the model operates with continuous or discrete tokens. This implies scalable behavior at the level of (a) generalization—generalizing to new data using the same metric as the training loss—which aligns with observations in language models (Kaplan et al., 2020). However, as for generalizing to a different metric, such as FID or GenEval score, although performance consistently improves with better validation loss, the trend may not follow a strict power-law.

Building on these findings, we scale the Fluid model, *i.e.*, random-order model with continuous tokens, up to 10.5B parameters and train it using the WebLI dataset (Chen et al., 2022). The resulting Fluid 10.5B model achieves a zero-shot FID of 6.16 on MS-COCO and a GenEval overall score of

0.69, comparing favorably with leading text-to-image generative models such as DALL-E 3 (Betker et al., 2023) and Stable Diffusion 3 (Esser et al., 2024). We hope that our empirical findings and positive results could shed light on the scaling behavior for text-to-image generation models and further innovation in this frontier.

## 2 RELATED WORK

**Text-to-image diffusion models.**    The dominant approaches are based on diffusion models. Dall-E 2 (Ramesh et al., 2022), Imagen (Ho et al., 2022), and Stable Diffusion (Rombach et al., 2022a) revolutionized text-to-image generation. Most recently, SD v3 (Esser et al., 2024) and Imagen3 (Baldridge et al., 2024) can generate realistic images that are hard to tell from real images. However, generating samples is usually computationally expensive due to the multiple forward passes.

**Autoregressive (AR) models.**    While being the defacto model for language modeling, AR models are lagging behind diffusion models for text-to-image generation. AR models (Yu et al., 2022; Chang et al., 2023; Li et al., 2023; Yu et al., 2023) are often used together with discrete tokenizers (van den Oord et al., 2017; Esser et al., 2021), which often limit the modeling capability. For example, Parti (Yu et al., 2022) scales up the model to 20B, which obtains a slightly lower/better FID score of 7.23 on MS-COCO than 7.27 by the 3.4B diffusion-based Imagen. Here we find that, with a continuous tokenizer, our small 369M model can already achieve the same FID score as Parti with 20B model.

Recently Li et al. (2024) challenges the conventional wisdom and replaces the discrete tokenizer with a continuous tokenizer via a diffusion loss. They introduced masked autoregressive (MAR) model obtains strong results for class-conditioning generation on ImageNet. However, scaling MAR models for text-to-image generation is unexplored. Here we empirically study the scaling behavior for MAR and report several findings important for the research community.

**Scaling language models.**    Kaplan et al. (2020) empirically observed that, for language model, the validation loss scales as a power-law with model size, dataset size, and the amount of compute used for training. Hoffmann et al. (2022) discovered that contemporary LLMs are under-trained, and that for compute-optimal training, the model size and the number of training tokens should be scaled equally. Under the same compute budget while using 4x more more data, their 70B Chinchilla outperforms much larger models, such as the 530B Megatron-Turing NLG (Smith et al., 2022). Wei et al. (2022) found that larger models have emergent abilities that are not present in smaller models. These observations have inspired significant efforts to scale up language models to trillions of parameters (Achiam et al., 2023; Team et al., 2023; Dubey et al., 2024).

**Scaling vision models.**    Similar scaling law has been obscure for computer vision. For recognition models (Tan, 2019; Zhai et al., 2022; He et al., 2022; Dehghani et al., 2023), scaling often comes with diminishing returns. For example, Dehghani et al. (2023) scale ViT up from 3B to 22B but only observed 0.25% accuracy increase in ImageNet linear probing. The scaling of generative models is more promising - DiT (Peebles & Xie, 2023) shows consistent improvement in generation quality when scaling up compute and model size (despite only up to 600M). The follow-up Sora (OpenAI, 2024) further shows the potential to scale up for video generation. In this paper, we perform a comprehensive empirical study of AR models for text-to-image generation.

## 3 PRELIMINARY: AUTOREGRESSIVE IMAGE GENERATION

Given a sequence of tokens $\{x^1, x^2, ..., x^n\}$ where the superscript $1 \leq i \leq n$ specifies an order, autoregressive models (Gregor et al., 2014; van den Oord et al., 2016b;a; Parmar et al., 2018; Chen et al., 2018; 2020) formulate the generation problem as "next token prediction":

$$p(x^1, ..., x^n) = \prod_{i=1}^{n} p(x^i \mid x^1, ..., x^{i-1}).\qquad(1)$$

Following the chain rule, the network is trained to model $p(x^i \mid x^1, ..., x^{i-1})$ and generate tokens iteratively. While all autoregressive models share this fundamental approach, differences in their

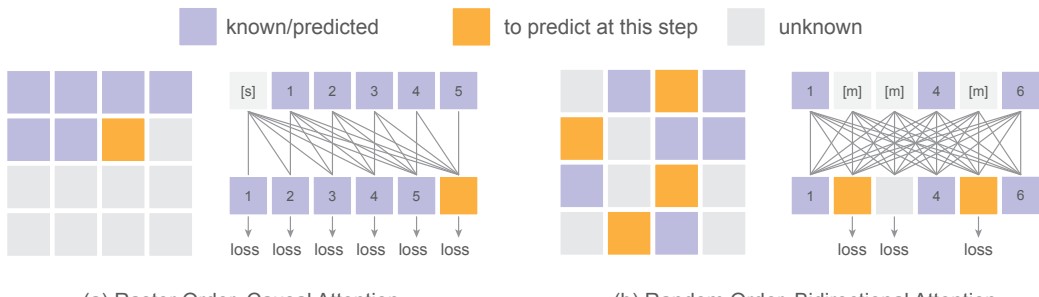

(a) Raster Order, Causal Attention  (b) Random Order, Bidirectional Attention

Figure 2: **Autoregressive models with different orders.** (a) A raster-order autoregressive model predicts one next token based on the known ones, implemented using a GPT-like transformer with causal attention. (b) A random-order autoregressive model predicts one or multiple tokens simultaneously given a random order, implemented using a BERT-like transformer with bidirectional attention.

design can affect the performance. Two key design choices are the representation of $x$, *i.e.*, discrete or continuous, and the generation order, which we elaborate below.

**Discrete *vs*. continuous tokens.** The goal of an autoregressive model is to estimate $p(x^i \mid x^1, ..., x^{i-1})$. Traditionally, this is done by transforming the image into a set of discrete tokens with a finite vocabulary and then estimating a categorical distribution over the vocabulary. The training objective is to minimize the cross-entropy loss, and sampling can be efficiently performed using categorical sampling. Most autoregressive image generation models rely on this form of token discretization (Esser et al., 2021; Chang et al., 2022; Tian et al., 2024; Yu et al., 2022).

However, such discretization often leads to a significant loss of information from the image (Figure 4). Recent work (Li et al., 2024) has shown the possibility of applying a small diffusion model to approximate the distribution of each image token in a continuous fashion. This approach eliminates the need for vector quantization, and allows modeling images with continuous tokenizers which yield much better reconstruction visual quality. In this paper, we explore the scaling behavior of autoregressive image models on both discrete and continuous tokens.

**Raster Order + GPT vs. Random Order + BERT.** In autoregressive image generation, there are two primary generation orders: raster and random. As illustrated in Figure 2, raster order generates tokens sequentially from left to right, top to bottom. This fixed-order generation is well-suited for a GPT-like transformer architecture, which predicts the next token in a causal manner. In contrast, random order allows multiple tokens to be generated in each step. The selection of these tokens can either be completely random or based on a sampling mechanism that prioritizes tokens with higher predicted confidence scores (Chang et al. (2023); Li et al. (2024)).

Each generation order has its pros and cons. Raster order models with GPT-like transformer support fast inference via key-value (kv) caching. However, this causal structure can also introduce performance degradation. On the other hand, random order generation is usually achieved with a BERT-like bidirectional attention mechanism. While this approach prevents the usage of kv-caching, it enables the model to decode multiple tokens at each autoregressive step, allowing global editing. Despite their individual strengths, it remains unclear in the literature which generation order scales better for text-to-image generation tasks. In this work, we compare the performance and scaling behaviors of raster-order and random-order autoregressive models.

## 4 Implementation

The overall framework of our text-to-image model training is straightforward. An image tokenizer first converts the original image into tokens. These tokens are then partially masked, and a transformer is trained to reconstruct the masked tokens conditioned on the text. Below, we provide a detailed description of each component of our framework (shown in Figure 3).

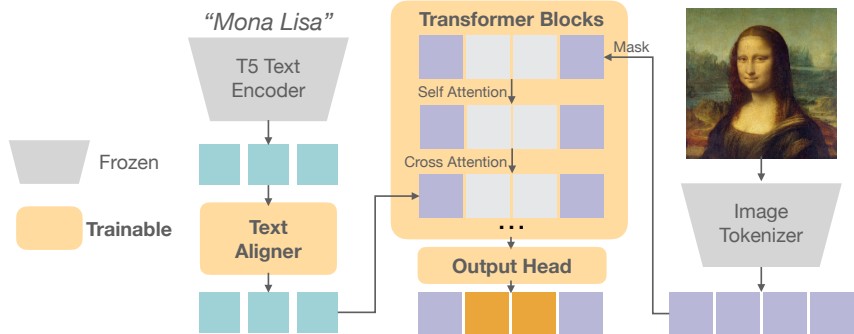

Figure 3: **Our text-to-image generation framework.** A pre-trained image tokenizer converts the image into either discrete or continuous tokens. The text is embedded using a pre-trained T5 encoder, followed by a trainable text aligner. The transformer then takes cross-attention from the text embeddings to predict the missing tokens (only random order model is shown here).

**Image Tokenizer.** We use a pre-trained image tokenizer to encode 256×256 images into a token space. Such a tokenizer can be either discrete or continuous, facilitating different training objectives of the autoregressive model. In our experiments, the discrete tokenizer is an VQGAN model (Esser et al., 2021) pre-trained on the WebLI dataset (Chen et al., 2022). We follow Muse (Chang et al., 2023) to encode each image into 16×16 discrete tokens with a vocabulary size of 8192. For the continuous tokenizer, we

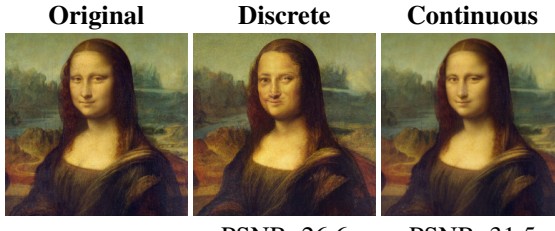

Figure 4: **Reconstruction quality of the tokenizers.** Image resolution is 256x256. The discrete tokenizer is significantly worse than the continuous tokenizer.

adopt a widely-used one from Stable Diffusion (Rombach et al., 2022b), which encodes the image into 32×32 continuous tokens, each containing 4 channels. To be consistent in sequence length with the discrete tokenizer, we further group each 2×2 patch of continuous tokens into a single token, resulting in a final sequence length of 256, with each token containing 16 channels. As shown in Figure 4, the continuous tokenizer can achieve notably higher reconstruction quality than the discrete one.

**Text Encoder.** The raw text (maximum length of 128) is tokenized by SentencePiece (Kudo, 2018), and embedded through a pre-trained T5-XXL encoder (Raffel et al., 2020), which has 4.7B parameters and is frozen during training. To further align the text embeddings for image generation, we add a small text aligner consisting of six trainable transformer blocks on top of the T5 embeddings, to extract the final text representation.

**Transformer.** After encoding the original image into a sequence of tokens, we use a standard decoder-only transformer model (Vaswani et al., 2017) for autoregressive generation. Each block consists of three consecutive layers – self-attention, cross-attention, and MLP. The self-attention and MLP layers are only applied to visual tokens, while the cross attention layer takes visual and textual tokens as queries and keys, respectively. As shown in Figure 2, for raster-order models, the transformer predicts the next token based on previous tokens using causal attention for the self-attention block, similar to GPT. In random-order models, unknown tokens are masked by a learnable token, and the transformer predicts these masked tokens using bidirectional attention, similar to BERT.

**Output head.** For *discrete* tokens, we follow the common practice with autoregressive models. The outputs are transformed into categorical distributions by softmax following a linear layer, whose weights are reused from the input embedding layer. For *continuous* tokens, we apply a six layer light-weight MLP as the diffusion head (Li et al., 2024) to model the per-token distribution. The embedding dimension of this head is the same as the backbone transformer. The per-token diffusion process follows (Nichol & Dhariwal, 2021; Li et al., 2024). The noise schedule has a cosine shape, with 1000 steps at training time; at inference time, it is resampled to 100 steps.

## 5 EXPERIMENTS

**Dataset.** We use a subset of the WebLI (Web Language Image) dataset (Chen et al., 2022) as our training set, which consists of image-text pairs from the web with high scores for both image quality and alt-text relevance. By default, the images are center-cropped and resized to 256×256.

**Training.** Unless otherwise specified, we use the AdamW optimizer ($\beta_1 = 0.9, \beta_2 = 0.95$) (Loshchilov & Hutter, 2019) with a weight decay of 0.02 to train each model for 1M steps with a batch size of 2048. This is equivalent to approximately 3 epochs on our dataset. For continuous tokens, we employ a constant learning rate schedule with a 65K-step linear warmup and a maximum learning rate of $1 \times 10^{-4}$; for discrete tokens, we use a cosine learning rate schedule as we find it to be better. For training the random-order models, we randomly sample the masking ratio from [0, 1] following a cosine schedule, similar to MaskGIT (Chang et al., 2022), to mask each image. For all models, exponential moving average of the weights are gathered by a decay rate of 0.9999 and then used for evaluation.

**Inference.** We follow the practices established by Imagen (Saharia et al., 2022), Muse (Chang et al., 2023), and Parti (Yu et al., 2022) to generate images from text prompts without rejection sampling. For random-order models, we use 64 steps for generation with a cosine schedule (Chang et al., 2022). To further enhance generation performance, we apply temperature and classifier-free guidance, as is commonly practiced.

**Evaluation.** We evaluate the scaling behavior of different autoregressive model variants both quantitatively and qualitatively. Quantitatively, we evaluate the validation loss on 30K images from the MS-COCO 2014 training set, as well as two widely-adopted metrics: zero-shot Frechet Inception Distance (FID) on MS-COCO, and the GenEval score (Ghosh et al., 2024). Inference hyper-parameters, such as temperature and classifier-free guidance, are optimized for each evaluation metric. FID is computed over 30K randomly selected image-text pairs from the MS-COCO 2014 training set, providing a metric that evaluates both the fidelity and diversity of generated images. The GenEval benchmark, on the other hand, measures the model's ability to generate images that accurately reflect the given prompt. For qualitative evaluation, we generate images from several prompts using each model and compare the visual quality of the generated images.

### 5.1 SCALING BEHAVIORS

In this section, we explore how two key design choices in autoregressive image generative models—token representation and generation order—affect performance and scaling behavior. We construct models with different combinations of these two design choices, resulting in four distinct variants of autoregressive image generation models. We also explore the generalizability of these models across different data and evaluation metrics. Our experiments reveal several intriguing properties.

**Validation losses consistently scale with model size.** In Figure 5, we examine the scaling behavior of the four autoregressive variants in terms of validation loss. We observe a linear relationship between validation loss and model size in the log space, as we increases model size from 150 million to 3 billion parameters. This aligns with the power-law finding in Henighan et al. (2020). This demonstrates that the improvements in training loss resulting from increased model size generalize well to validation loss on data different from the training data.

**Random-order models with continuous tokens scale the best in evaluation scores.** In Figure 6, we analyze the scaling behavior of the four autoregressive variants in terms of FID and GenEval overall scores. We find that the improvements observed in validation loss do not always translate linearly to better evaluation metrics, implying that there is no strict power-law relationship between these metrics and model size. For example, raster-order models with discrete tokens (blue line) reach a plateau in both FID and GenEval scores around 1B parameters. Among the four variants, random-order models with continuous tokens (*i.e.*, Fluid) show consistent improvements in evaluation metrics up to 3B parameters, achieving the best overall performance. Therefore, we focus on further investigating the scaling behavior of this model.

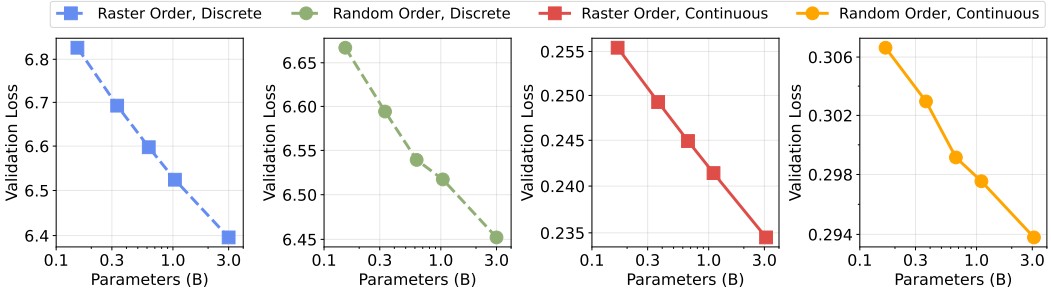

Figure 5: **Validation loss scales as a power-law with model size.** The validation loss is evaluated on 30K images randomly sampled from the MS-COCO 2014 training set. The x and y axes are in log-scale. The change in y is relatively small for each plot, making the log-scale alike linear-scale.

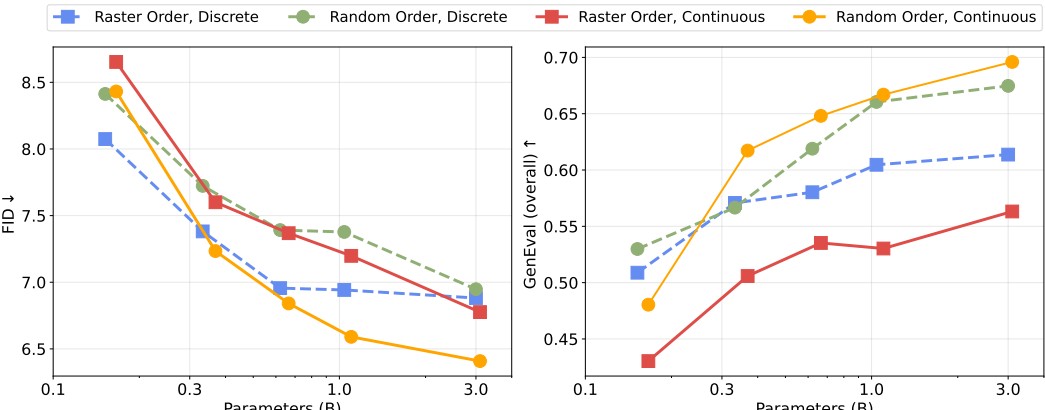

Figure 6: **Random-order models using continuous tokens (orange) achieve the best performance on evaluation metrics.** FID (lower is better) is evaluated on 30K images randomly sampled from the MS-COCO 2014 training set, while the GenEval overall score (higher is better) is assessed using the 553 prompts provided by the official benchmark, with four images generated for each prompt. Among all models, random-order models on continuous tokens consistently show an improvement in evaluation metrics as model size increases and achieve the best FID and GenEval scores.

**Random-order models with continuous tokens scale with training computes.** In Figure 7, we plot validation loss, FID, and GenEval scores as functions of total training steps and compute for different model sizes of Fluid. We observe consistent improvements in both validation loss and evaluation performance with increased training steps and compute. However, the benefits from additional training steps saturate around 1M steps, indicating that training smaller models for more steps is less compute-efficient compared to training larger models for fewer steps. This behavior aligns with observations in language models, highlighting the potential for scaling up model sizes with sufficient training.

**Strong correlation between validation loss and evaluation metrics.** In Figure 8, we plot FID and GenEval scores against validation loss for different model sizes of Fluid and observe a strong correlation. To quantify this, we fit the data points using linear regression. The Pearson correlation coefficients for FID and GenEval scores are 0.917 and -0.931, respectively, indicating a nearly linear relationship between validation loss and these evaluation metrics across model sizes ranging from 150M to 3B[1]. Encouraged by this positive trend, we trained a model with 10.5B parameters and a batch size of 4096 for 1M steps, achieving state-of-the-art text-to-image generation performance, as discussed in the next section.

---

[1]Since both FID and GenEval scores have lower/upper bounds, this linear relationship cannot hold indefinitely. Future work should explore the limits of this correlation.

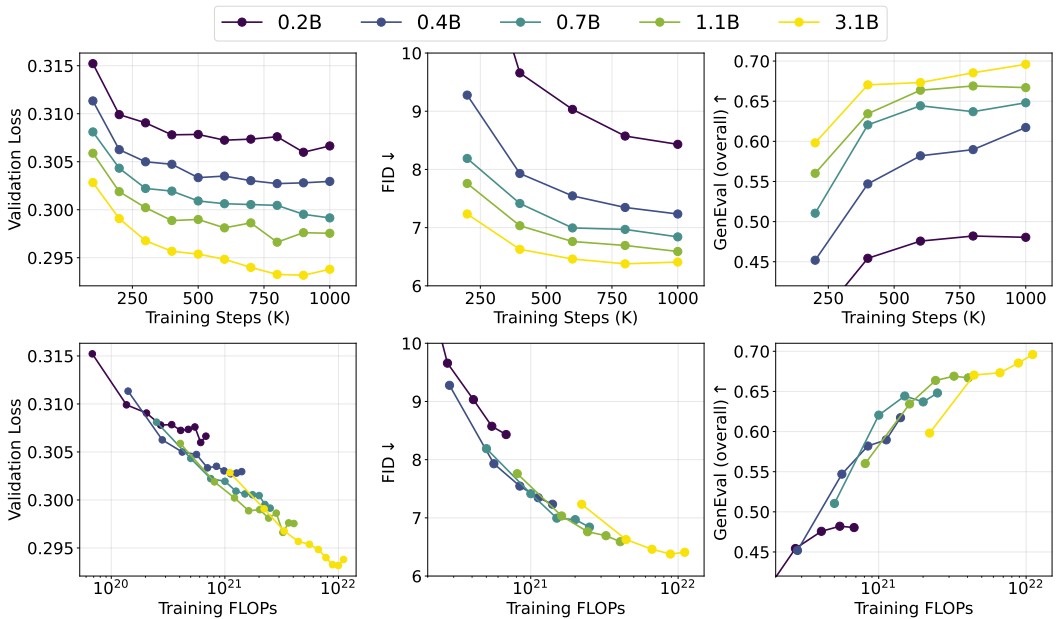

Figure 7: **Validation losses and evaluation performance scale with increasing training steps and computes.** We use random-order models with continuous tokens. Results for other autoregressive variants are included in the appendix. The training compute is computed as model GFLOPs×batch size×training steps×3, where the factor of 3 accounts for the backward pass being approximately twice as compute-intensive as the forward pass.

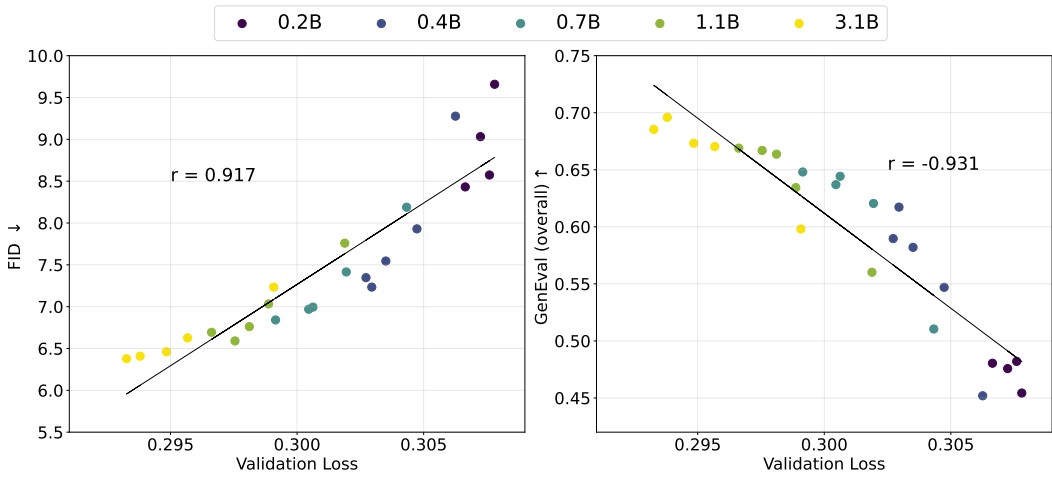

Figure 8: **Validation loss and evaluation metrics are highly correlated.** We use random-order models with continuous tokens. The Pearson correlation coefficients for FID and GenEval scores are 0.917 and -0.931, respectively. We also observe that the linear correlation slightly weakens and becomes less pronounced for the 3.1B model.

**Continuous tokens and large models are crucial for visual quality.** In Figure 9, we compare the visual quality of images generated by the four autoregressive variants. The visual quality of models using discrete tokens is significantly worse than that of models using continuous tokens, *e.g.*, the eyes of the corgi is asymmetric for discrete token based models and scaling up can not solve this problem. This limitation is largely because of the discrete tokenizer, which introduces substantial information loss. For instance, even with 3B parameters, the discrete token models cannot generate

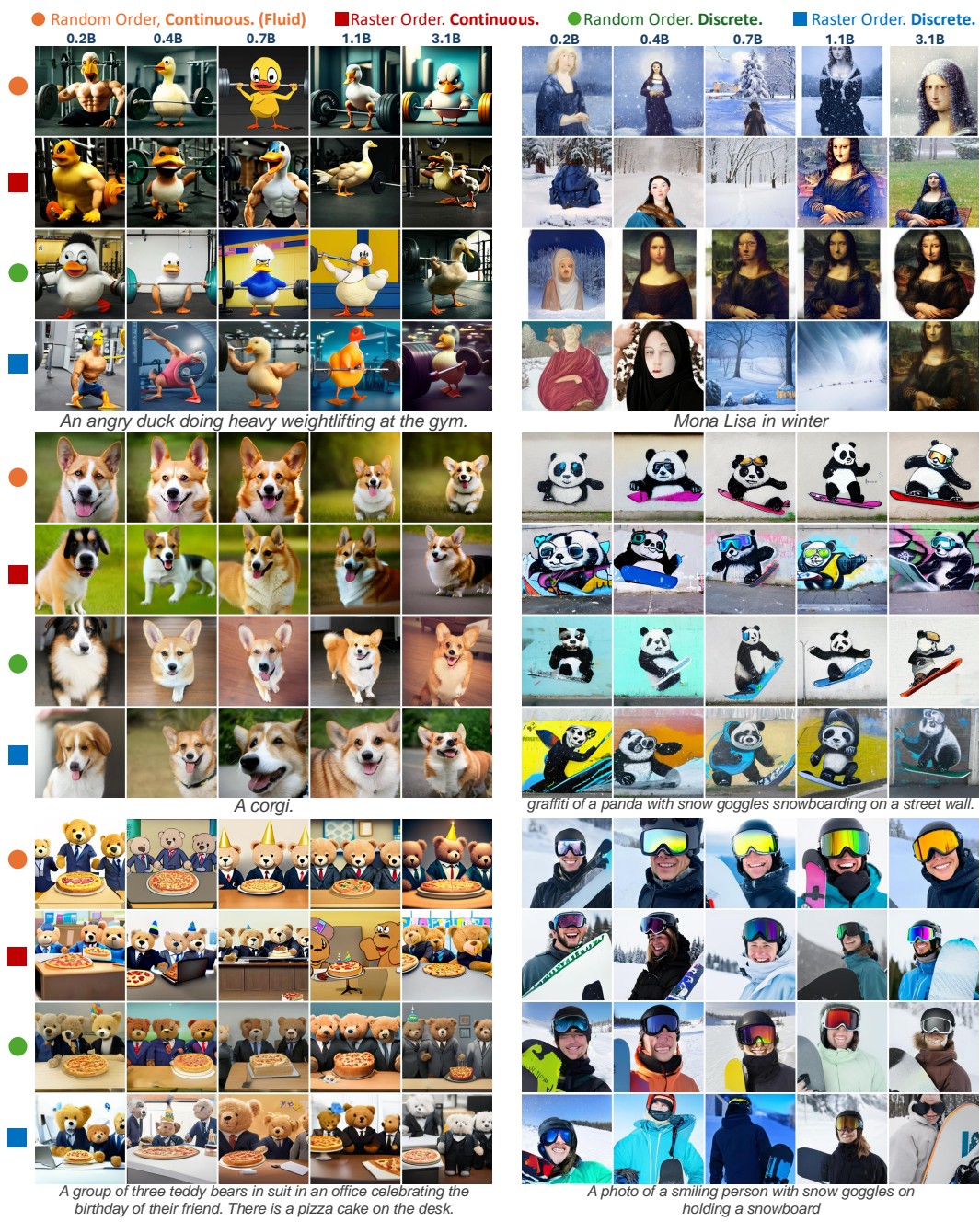

Figure 9: **Visual quality and image-text alignment improves with increasing model size.** *Best viewed zoomed-in.* Fluid 🟠 achieves the highest visual quality and best image-text alignment.

an accurate Mona Lisa due to poor reconstruction quality of the tokenizer (Figure 4). In contrast, models with continuous tokens produce much higher-quality images.

Additionally, larger models show consistent improvements in both visual quality and image-text alignment. For example, a random-order model with 0.2B parameters struggles to generate "an angry duck doing heavy weightlifting at the gym", while the same model with 3B parameters can generate the corresponding images successfully. This demonstrates that modeling continuous tokens and increasing model size are crucial for achieving high visual quality in autoregressive image generation models.

Table 1: **System-level comparison.** Fluid achieves leading results on both MS-COCO zero-shot FID-30K and GenEval benchmark (Ghosh et al., 2024). †: CM3Leon result is reported without retrieval.

| | #params | MS-COCO FID-30K↓ | Single Obj. | Two Obj. | Counting | Colors | Position | Color Attri. | Overall |
|---|---|---|---|---|---|---|---|---|---|
| *diffusion model* | | | | | | | | | |
| LDM | 1.4B | 12.64 | 0.92 | 0.29 | 0.23 | 0.70 | 0.02 | 0.05 | 0.37 |
| DALL-E 2 | 4.2B | 10.39 | 0.94 | 0.66 | 0.49 | 0.77 | 0.10 | 0.19 | 0.52 |
| DALL-E 3 | - | - | 0.96 | 0.87 | 0.47 | 0.83 | 0.43 | 0.45 | 0.67 |
| Imagen | 3B | 7.27 | - | - | - | - | - | - | - |
| SD3 | 8B | - | 0.98 | 0.84 | 0.66 | 0.74 | 0.40 | 0.43 | 0.68 |
| Transfusion | 7.3B | 6.78 | - | - | - | - | - | - | 0.63 |
| RAPHAEL | 3B | 6.61 | - | - | - | - | - | - | - |
| *autoregressive model* | | | | | | | | | |
| CM3Leon† | 7B | 10.82 | - | - | - | - | - | - | - |
| Show-o | 1.3B | 9.24 | 0.95 | 0.52 | 0.49 | 0.82 | 0.11 | 0.28 | 0.53 |
| Muse | 3B | 7.88 | - | - | - | - | - | - | - |
| Parti | 20B | 7.23 | - | - | - | - | - | - | - |
| **Fluid (our work)** | 369M | 7.23 | 0.96 | 0.64 | 0.53 | 0.78 | 0.33 | 0.46 | 0.62 |
| | 665M | 6.84 | 0.96 | 0.73 | 0.51 | 0.77 | 0.42 | 0.51 | 0.65 |
| | 1.1B | 6.59 | 0.96 | 0.77 | 0.61 | 0.78 | 0.34 | 0.53 | 0.67 |
| | 3.1B | 6.41 | 0.98 | 0.83 | 0.60 | 0.82 | 0.41 | 0.53 | **0.70** |
| | 10.5B | **6.16** | 0.96 | 0.83 | 0.63 | 0.80 | 0.39 | 0.51 | 0.69 |

## 5.2 BENCHMARKING WITH PREVIOUS SYSTEMS

In this section, we compare our Fluid, *i.e.*, continuous random-order autoregressive model, with leading text-to-image generation systems in Table 1 (Rombach et al., 2022b; Ramesh et al., 2022; Betker et al., 2023; Saharia et al., 2022; Esser et al., 2024; Zhou et al., 2024; Xue et al., 2024; Yu et al., 2023; Xie et al., 2024; Chang et al., 2023; Yu et al., 2022). The Fluid smallest model, with 369M parameters, achieves a zero-shot FID of 7.23 on MS-COCO and a GenEval overall score of 0.62, matching the performance of many state-of-the-art models with several billion parameters (*e.g.*, Parti with 20B parameters only achieves 7.23). The Fluid largest model, with 10.5B parameters, further improves the zero-shot FID on MS-COCO to 6.16 and increases the GenEval overall score to 0.69[2], with a speed of 1.571 seconds per image per TPU (evaluated on 32 TPU v5 with a batch size of 2048). Detailed model configurations and generation speed results are included in the appendix. We hope these strong results and promising scaling behavior provide valuable insights and support for the scalability of autoregressive models in visual generative modeling.

## 6 DISCUSSION

In this paper, we present an empirical study on the scaling behavior of autoregressive models for text-to-image generation. We investigate two critical design factors: random order versus raster order, and discrete tokens versus continuous tokens. Our results show that random-order models with continuous tokens achieve the best performance and scaling behavior across various evaluation metrics and in terms of visual quality. Building on these findings, we scale up the random-order model with continuous tokens, namely Fluid, to 10.5B parameters, and achieves state-of-the-art text-to-image generation performance. We hope that our findings and promising results could provide valuable insights into the scaling behavior of autoregressive models for image generation and help bridge the gap between the scaling performance of vision models and language models.

**Reproducibility Statement.** To aid reproducibility, we have provided the implementation details of our framework in Section 4, training hyper-parameters in Section 5, and model configurations in the appendix. For the diffusion loss used for continuous tokens, we have strictly followed the open-sourced code of Li et al. (2024).

**Acknowledgements.** We would like to express our gratitude to Amy Shen and Alex Rizkowsky for their assistance in securing computational resources, and to Yash Katariya, Ivy Zheng, and Roy Frostig for their valuable insights on JAX-related questions. We also thank Matan Cohen for his support with precomputed T5 embeddings. Our appreciation extends to Han Zhang, Jason Baldridge, Dilip Krishnan, David Salesin and VisCam team for their helpful discussions and valuable feedback.

---

[2]We observe that the GenEval scores plateau for the 10.5B Fluid compared to the 3.1B Fluid; however, it continues to show consistent improvements in visual quality and FID.

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
