# A    MORE SCALING RESULTS

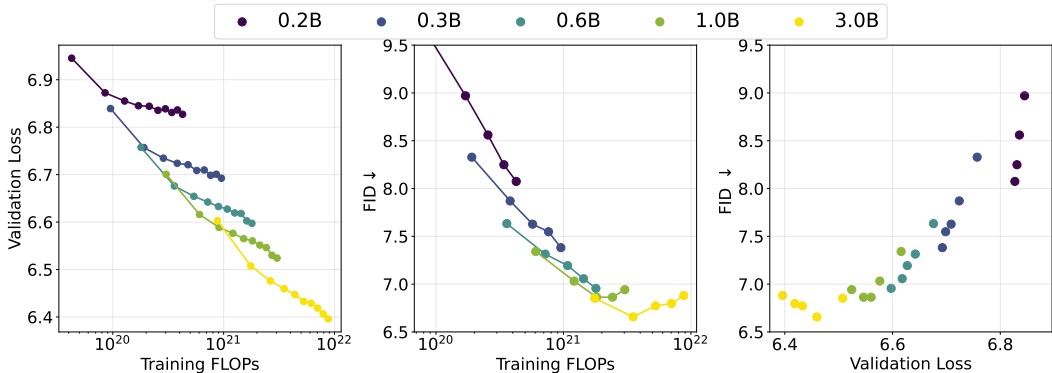

Figure A1: Validation loss and FID w.r.t. training FLOPs for raster-order models with discrete tokens.

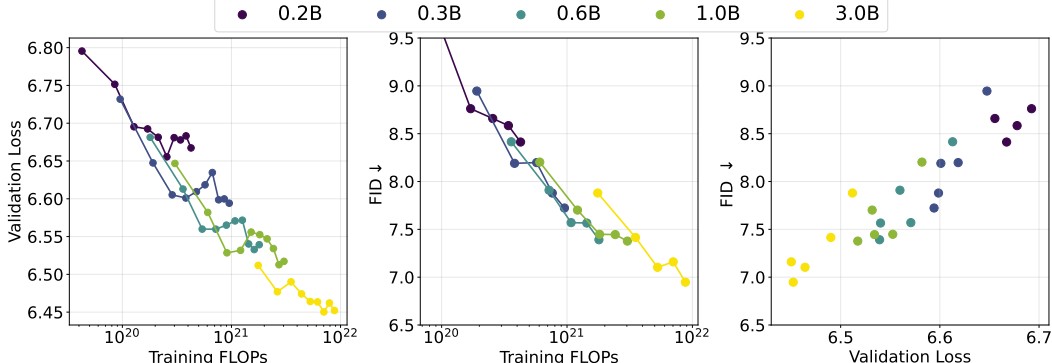

Figure A2: Validation loss and FID w.r.t. training FLOPs for random-order models with discrete tokens.

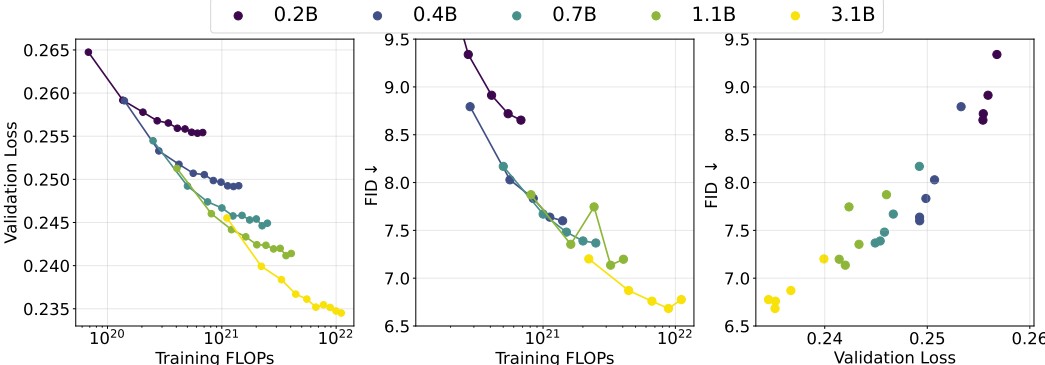

Figure A3: Validation loss and FID w.r.t. training FLOPs for raster-order models with continuous tokens.

## A.1    SCALING BEHAVIOR W.R.T TRAINING FLOPS.

In Figure A1 − A3, we present the relationship between validation loss and FID with respect to training FLOPs for the other three autoregressive variants. As shown, all three variants exhibit consistent scaling behavior in validation loss, but the FID gains start to level off for the 3B raster-order model that uses discrete tokens. This hints that simply using a GPT-like language model for images in a straightforward way may not scale well. To improve scaling for visual data, further adaptations such as continuous tokens and random-order generation are needed.

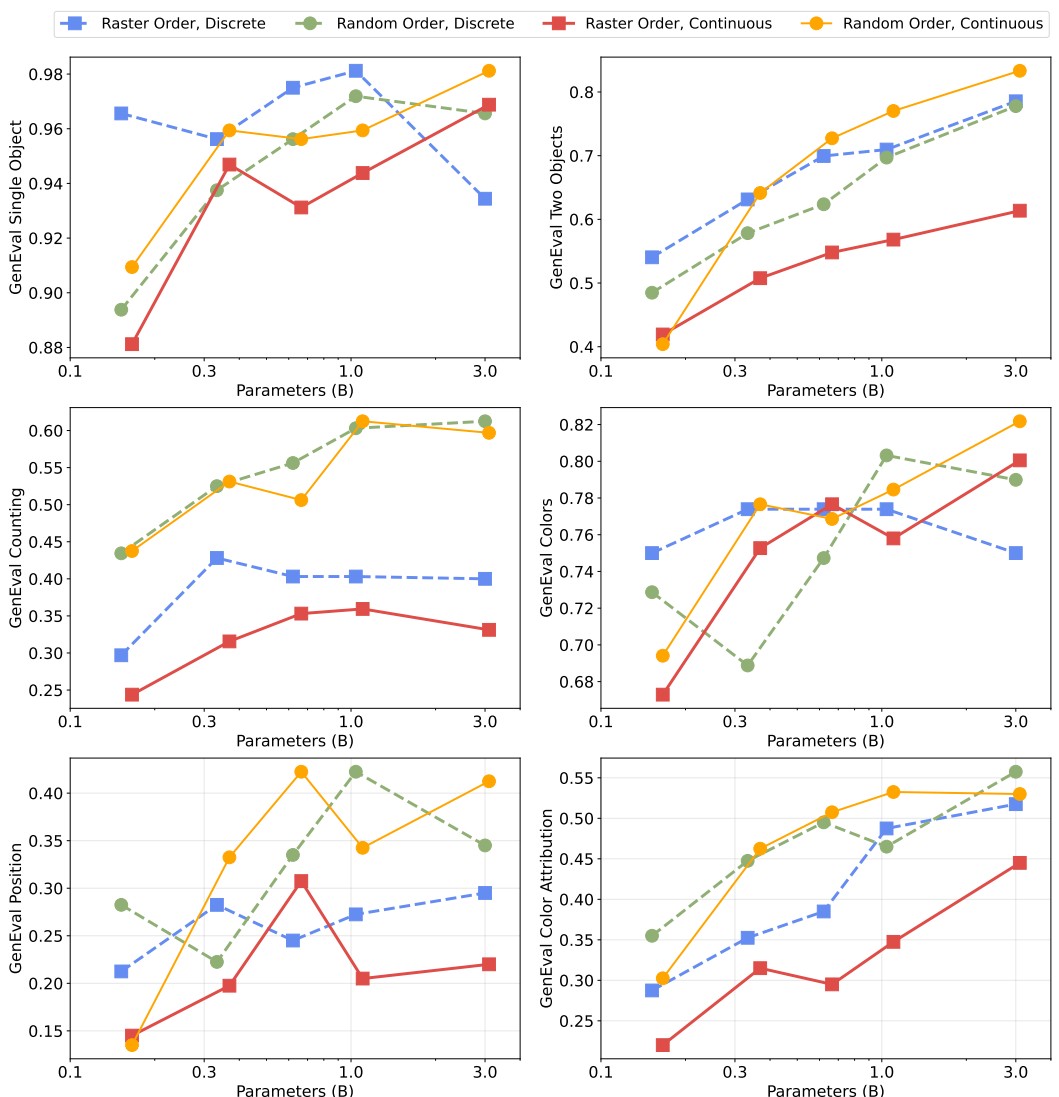

Figure A4: Scaling behavior on fine-grained GenEval scores across different model setups.

## A.2 Fine-grained GenEval scores.

In Figure A4, we present the performance of the four autoregressive variants across all metrics in the GenEval benchmark. As shown, all models perform well in single-object scenarios. The performance on other metrics also consistently improves as model size increases. Additionally, we observe that random-order models significantly outperform raster-order models in metrics related to counting and position, both of which require a better global generation structure—an area where random-order models have an advantage.

## A.3 Scaling comparison with SD3

The scaling of diffusion-based models, and later flow-based models, has been studied in several works in the literature. Most notably, in the SD3 paper (Esser et al., 2024) (one of the current state-of-the-art flow-based models), they scaled the model from 0.8B to 8B parameters and observed consistent scaling behavior. In Figure A5, we compare the scaling behavior of SD3 and Fluid in terms of the GenEval score. Surprisingly, despite various differences in model architecture, training data, and implementations, the scaling rates of these two models are quite similar within the same GenEval score range (0.6–0.7). Moreover, Fluid performs even better than SD3 at the same number of parameters. However, we also observe that the GenEval scores plateau for the 10.5B Fluid model

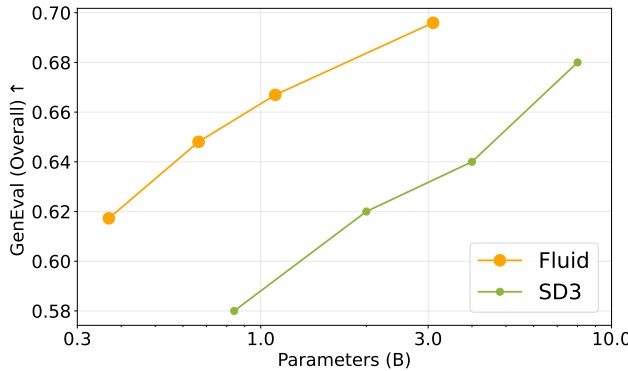

Figure A5: Scaling comparison between our Fluid model and SD3 across different backbone sizes on GenEval benchmark.

compared to the 3.1B Fluid model. We hypothesize that this is due to the inconsistency between the scaling of the validation loss and the validation metrics. The scaling behavior of SD3 beyond 10B parameters would also be interesting to explore, and we leave it for future work.

## B    MORE IMPLEMENTATION DETAILS

**Model Configurations.** In Table A1, we provide the detailed configurations of our models across different sizes. The MLP ratio is fixed at 4 for all models. The text aligner consistently consists of 6 transformer blocks, with the same channel size as the image transformer. The DiffLoss MLP also contains 6 MLP layers, with channels matching those of the image transformer. The generation speed is

Table A1: Model configurations of our random-order models on continuous tokens.

| #Params | #Blocks | #Channels | #Heads | Speed (sec/img) |
|---|---|---|---|---|
| 166M | 12 | 768 | 12 | 0.047 |
| 369M | 16 | 1024 | 16 | 0.078 |
| 665M | 20 | 1280 | 16 | 0.110 |
| 1.1B | 24 | 1536 | 16 | 0.180 |
| 3.1B | 32 | 2304 | 24 | 0.483 |
| 10.5B | 34 | 4096 | 64 | 1.571 |

evaluated on 32 TPU v5 with a batch size of 2048, and we report the time needed to generate one image per TPU.

**CFG and Temperature.**   To enable CFG (Ho & Salimans, 2022), we randomly replace the text condition with a dummy vacant text string for 10% of the samples. During inference, the model is run with the given text condition and the vacant text, yielding two outputs for each token.

For discrete tokens, the two outputs are conditional logit $\ell_c$ and unconditional logit $\ell_u$. Then the final logit $\ell_g$ with a guidance scale of $\omega$ is $\ell_g = (1 + \omega) \cdot \ell_c - \omega \cdot \ell_u$. Afterwards, the final logit is divided by the temperature $\tau$, which controls the diversity of the samples.

For continuous tokens, they are conditional vector $z_c$ and unconditional vector $z_u$ for the diffusion loss head. The predicted noise $\epsilon$ is then extrapolated as: $\epsilon = \epsilon_\theta(x_t|t, z_u) + \omega \cdot (\epsilon_\theta(x_t|t, z_c) - \epsilon_\theta(x_t|t, z_u))$, where $\omega$ is the guidance scale. To control sample diversity via temperature $\tau$, Dhariwal & Nichol (2021) suggests to either divide $\epsilon_\theta$ by $\tau$ or scale the noise with $\tau$. We follow MAR (Li et al., 2024) to adopt the latter option.

Table A2: Optimal guidance scale $\omega$ and temperature $\tau$ for FID.

| Model variants | $\omega$ | $\tau$ |
|---|---|---|
| random order, continuous token | 5 | 0.975 |
| raster order, continuous token | 4.5 | 0.975 |
| random order, discrete token | 1.6 | 1.05 |
| raster order, discrete token | 2.5 | 0.95 |

We conduct a sweep over the guidance scale $\omega$ and temperature $\tau$ to determine the optimal combination for each model variant. This sweep is performed for models with 160M and 360M parameters, and we find that the optimal parameters remain consistent across these two scales. Therefore, we apply the same parameters, as shown in Table A2, for models with sizes up to 3B parameters.

**Visualization Details of Figure 1.** To generate Figure 1, we first used our 10.5B random-order model with continuous tokens to generate 256×256 images conditioned on the text prompts. We then applied an in-house super-resolution model to upscale the images to 1024×1024 for improved visual quality (only for Figure 1). The prompts used, from left to right and top to bottom, are: "Close up photo of a knight", "A baseball bat", "An origami bird made of paper is perched on a branch of an evergreen tree", "A wise old mushroom wearing spectacles and reading a book under a tree", "photo of an eagle with a golden crown resting upon its head", "A beautiful castle beside a waterfall in the woods by Josef Thoma, matte painting, trending on artstation HQ", "A photorealistic image of a beautiful mountain range reflected perfectly on the still surface of a crystal-clear lake, surrounded by lush green meadows and vibrant wildflowers", "Oil painting of a vibrant landscape of rolling hills covered in wildflowers, with a quaint farmhouse nestled in the distance under a bright blue sky", "Mona Lisa on a cake", "A grumpy-looking lemon wearing a raincoat and holding an umbrella, standing in the rain", "A hyperrealistic close-up of an eye, with the iris reflecting a vast and detailed landscape, complete with mountains, rivers, and forests", "A section of the Great Wall in the mountains, detailed charcoal sketch", "A present with a blue ribbon under a Christmas tree".

**Ablation study on Text Aligner.** Table A3 presents a pilot study on the effect of the trainable text aligner. We trained the smaller model with 277M parameters using random order continuous tokens with a T5-XL text encoder for 500k steps. The FID score shows improvements as more layers are added to the text aligner. We choose to use 6 layers in our Fluid models to balance performance and efficiency.

Table A3: FID vs number of layers in the text aligner.

| #layers | FID |
|---------|------|
| 0 | 9.38 |
| 3 | 8.61 |
| 6 | 8.42 |

## C  ADDITIONAL QUALITATIVE RESULTS

### C.1  IMAGES FROM 10.5B FLUID MODEL

In Figure A6 and A7 we show additional images generated from our 10.5B Fluid model.

### C.2  COMPARISON BETWEEN 3.1B AND 10.5B FLUID MODEL

We also show more qualitative comparisons between the images generated by the 3.1B and 10.5B Fluidmodel in Figure A8 and A9. While the 3.1B model performs well in most cases, the 10.5B model demonstrates better ability in generating text and finer image details, and creating images that better align with the corresponding texts.

### C.3  COMPARISON BETWEEN FLUID ON 256x256 AND 512x512

To enhance image detail and evaluate our model's performance at higher resolutions, we trained a Fluid 3.1B model to generate 512x512 resolution images. This was achieved by replacing the learned positional embeddings with 2D Rotary Position Embeddings and initially training the model at 256x256 resolution. The 256x256 model served as the initialization for the 512x512 model, with the positional embeddings extended to support the higher resolution. Fine-tuning was conducted for 500k iterations at 512x512 resolution using a reduced learning rate of 1e-5 (0.1x the base learning rate). Figures A10, A11 and A12 compare the 512x512 Fluid 3.1B model with the original 256x256 Fluid 3.1B and 10.5B models, demonstrating that the 512x512 model generates sharper, more detailed images with significantly improved visual quality, while Fluid 10.5B model can generate better text and numbers.

### C.4  COMPARISON BETWEEN FLUID AND LLAMAGEN

Figures A13 and A14 present a qualitative comparison between our Fluid 3.1B and 10.5B models and LlamaGen (Sun et al. (2024)). LlamaGen employs a two-stage training process for text-to-

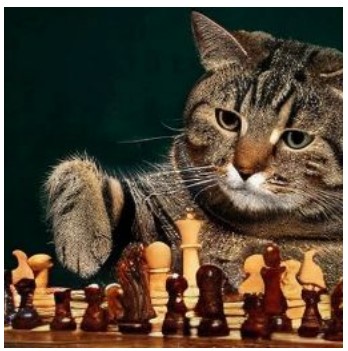

A photo of a cat playing chess.

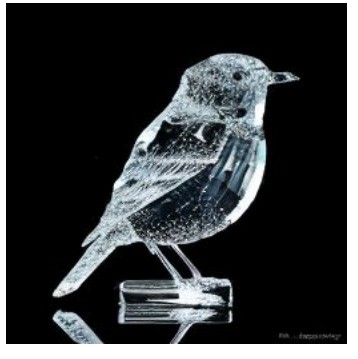

A bird made of crystal

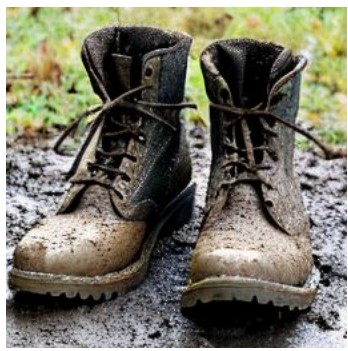

A pair of old boots covered in mud.

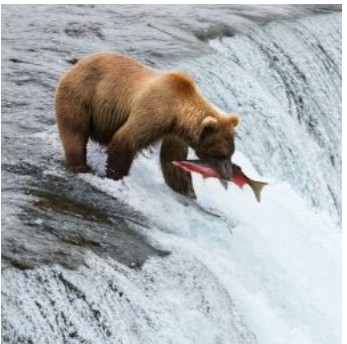

Photo of a bear catching salmon.

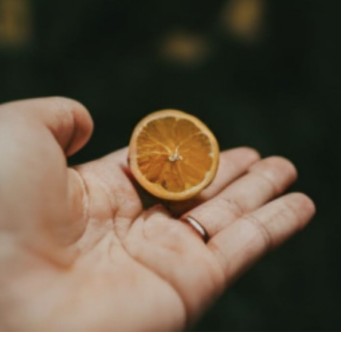

High quality, a close up photo of a human hand

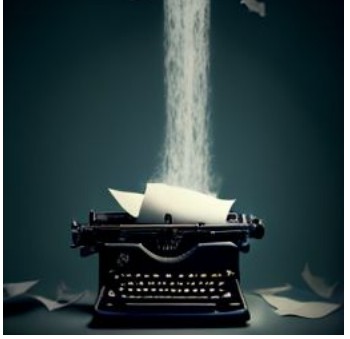

A vintage typewriter with paper spewing out like a waterfall.

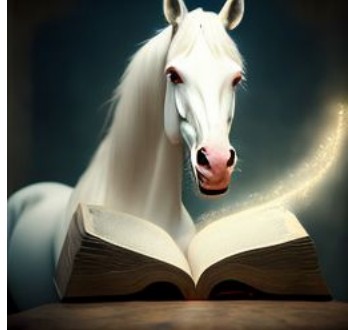

A white horse reading a book, fairytale.

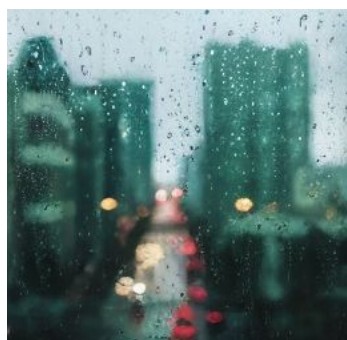

A window with raindrops trickling down, overlooking a blurry city.

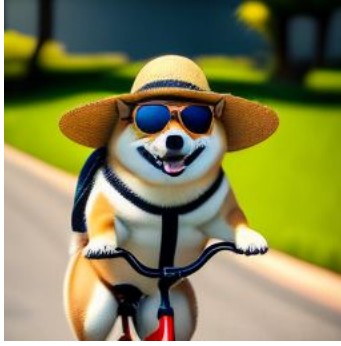

A photo of a Shiba Inu dog with a backpack riding a bike. It is wearing sunglasses and a beach hat.

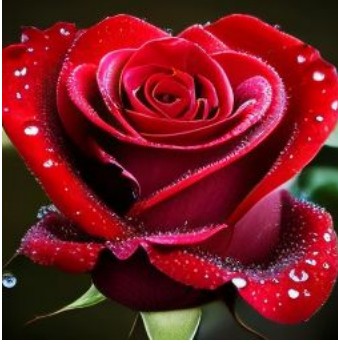

A close-up photo of a bright red rose, petals scattered with some water droplets, crystal clear.

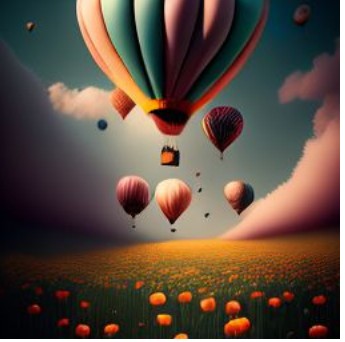

Hot air balloons and flowers, collage art, photorealism, muted colors, 3D shading beautiful eldritch, mixed media, vaporous

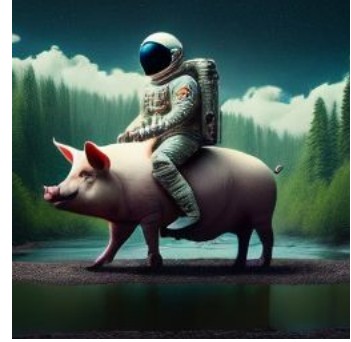

an astronaut rides a pig through in the forest. next to a river, with clouds in the sky

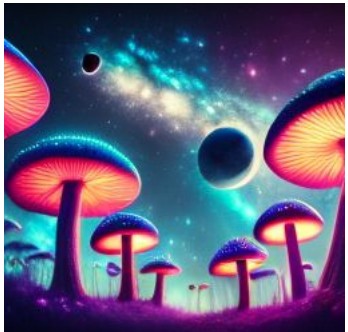

An otherworldly forest of giant glowing mushrooms under a vibrant night sky filled with distant planets and stars, creating a dreamlike, cosmic landscape

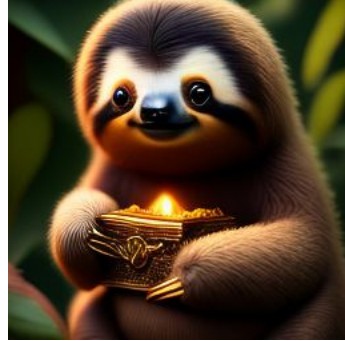

A close-up photo of a baby sloth holding a treasure chest. A warm, golden light emanates from within the chest, casting a soft glow on the sloth's fur and the surrounding rainforest foliage.

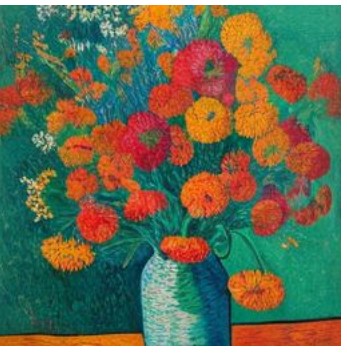

A still life of a vase overflowing with vibrant flowers, painted in bold colors and textured brushstrokes, reminiscent of van Gogh's iconic style.

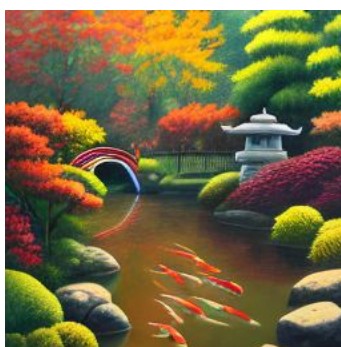

A tranquil scene of a Japanese garden with a koi pond, painted in delicate brushstrokes and a harmonious blend of warm and cool colors.

Figure A6: Additional images generated from our 10.5B Fluid model.

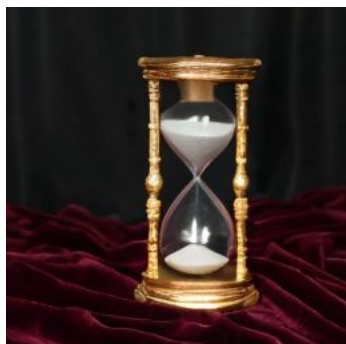

A golden hourglass, half-filled with flowing silver sand, placed on a rich velvet cloth.

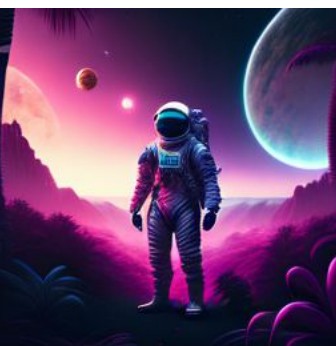

A space explorer discovering an alien jungle planet under a purple sky.

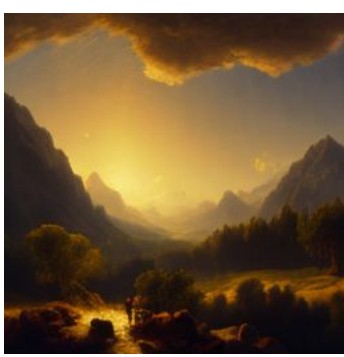

A gorgeous mountain landscape at sunset. Masterful painting by Rembrandt

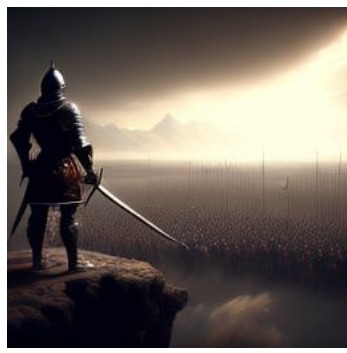

A medieval knight standing on a cliff overlooking a vast battlefield.

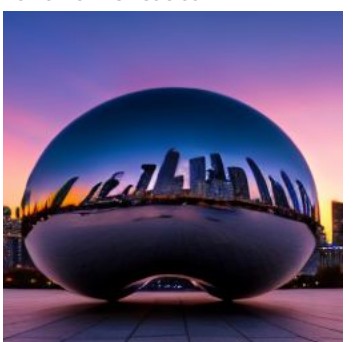

An image of a chrome sphere reflecting a vibrant city skyline at sunset.

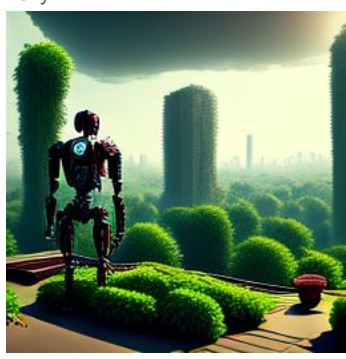

A post-apocalyptic city overtaken by vines, with a robot tending a rooftop garden.

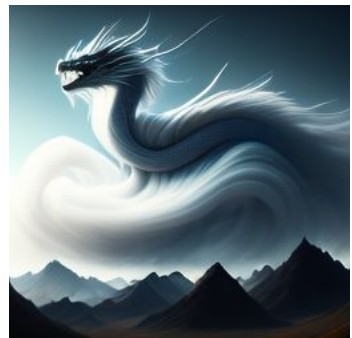

A cloud dragon flying over mountains, its body swirling with the wind

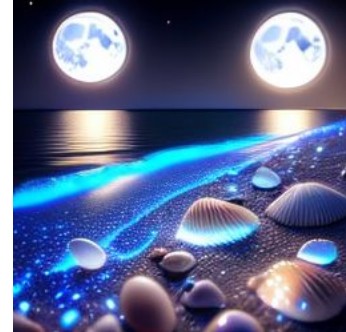

A moonlit beach with glowing seashells and two moons reflected on the water.

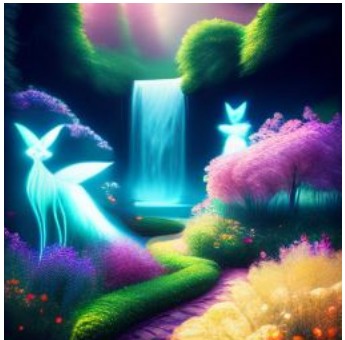

An enchanted garden where every plant glows softly, and creatures made of light and shadow flit between the trees, with a waterfall flowing in the background

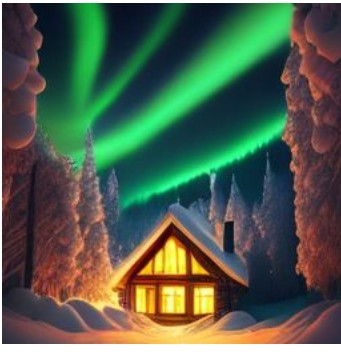

A cozy cabin in the middle of a snowy forest, surrounded by tall trees with lights glowing through the windows, a northern lights display visible in the sky.

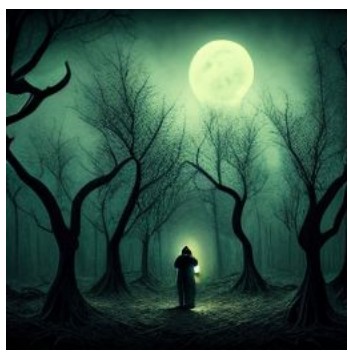

A dark forest under a full moon, with twisted, gnarled trees, shadows lurking behind every branch, and a lone figure holding a glowing lantern.

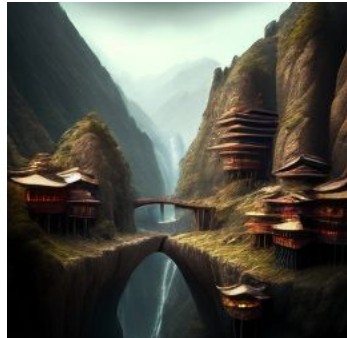

A mountain village built into the cliffs of a canyon, where bridges connect houses carved into rock, and waterfalls flow down into the valley below.

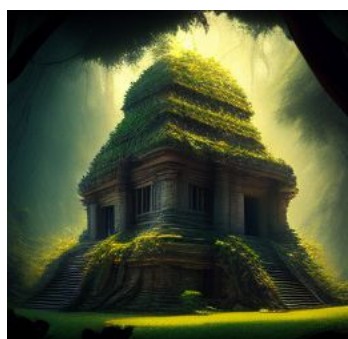

An ancient, overgrown temple hidden deep within a jungle, with vines crawling up its stone walls, and golden light filtering through the thick canopy above.

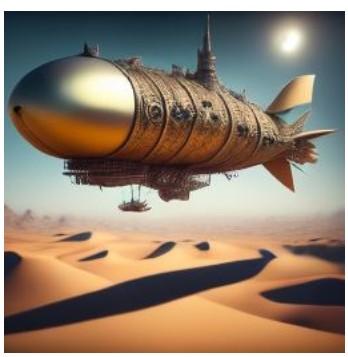

A steampunk airship soaring over a desert landscape, with mechanical wings and gears turning, casting shadows on the sand dunes below

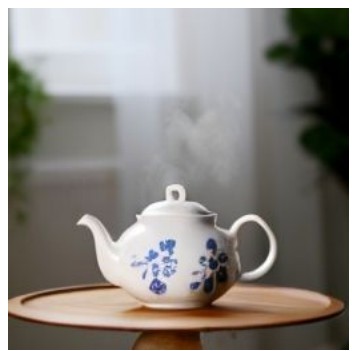

A pristine white teapot with delicate blue floral designs, steaming with hot tea, sitting on a round wooden table.

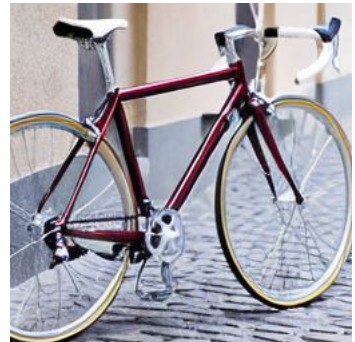

A sleek modern bicycle, leaning against a cobblestone alley, with shiny chrome details and a deep red paint job.

Figure A7: Additional images generated from our 10.5B Fluid model.

| **3.1B** | **10.5B** | **3.1B** | **10.5B** |
|---|---|---|---|

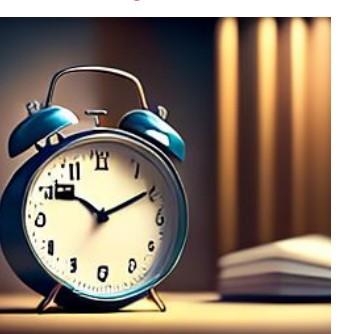 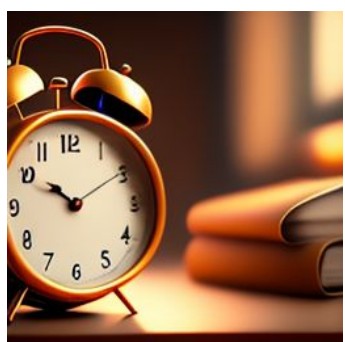 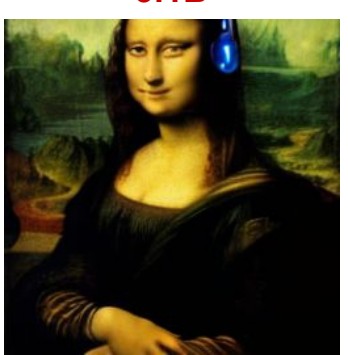 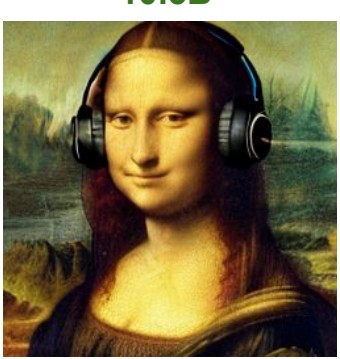

a clock on a desk, cartoon style       Mona Lisa wearing headphone

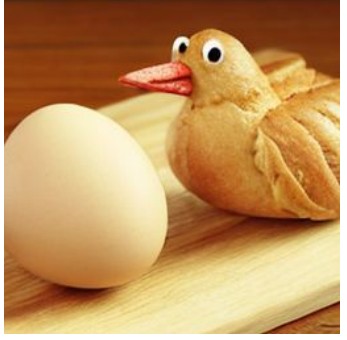 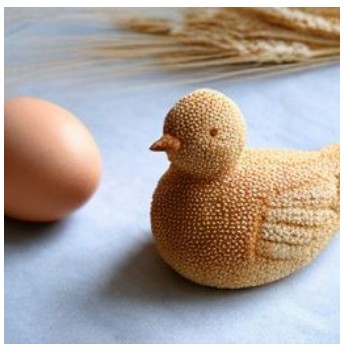 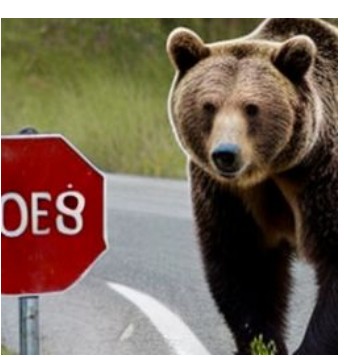 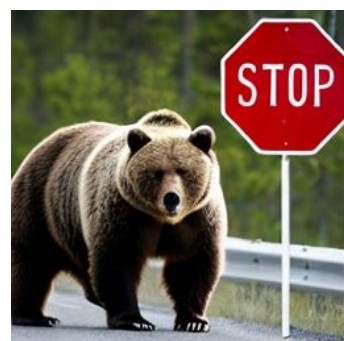

an egg and a bird made of wheat bread      a photo of a bear next to a STOP sign

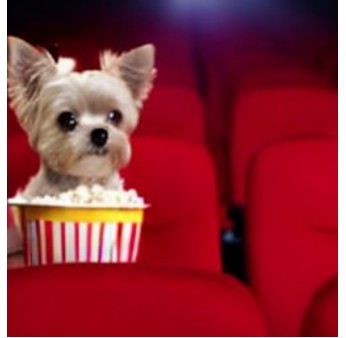 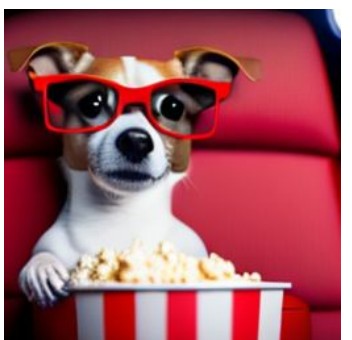 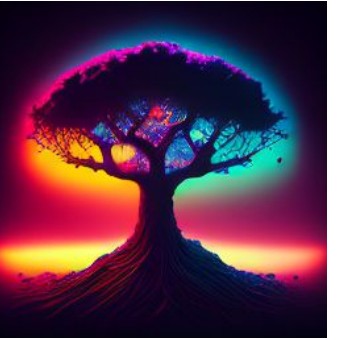 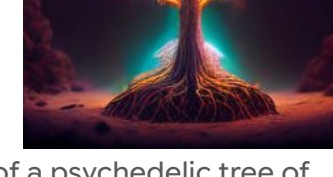

Cute small dog sitting in a movie theater eating popcorn watching a movie     dark high contrast render of a psychedelic tree of life illuminating dust in a mystical cave

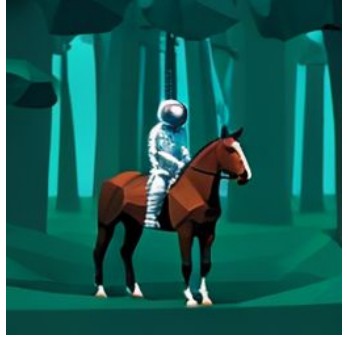 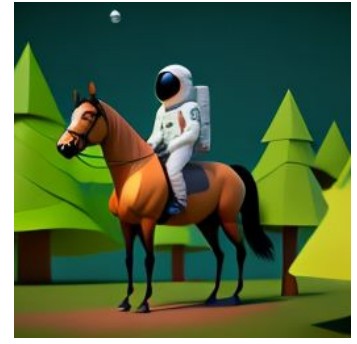 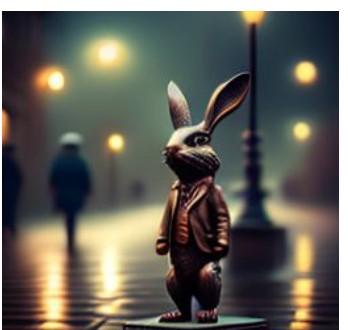 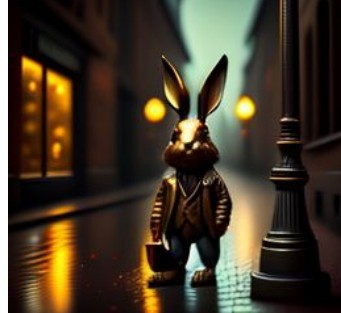

Astronaut riding a horse in a forest, low poly

Metal statue of a rabbit detective standing under a street light near a brick lined street on a rainy night. Bokeh

Figure A8: Additional qualitative comparisons between images generated from 3.1B and 10.5B Fluid model.

**3.1B** **10.5B** **3.1B** **10.5B**

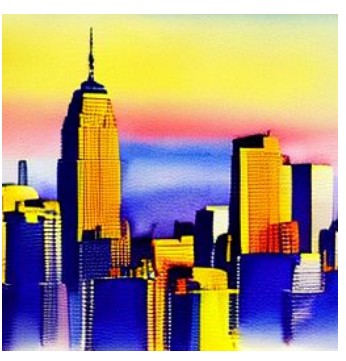 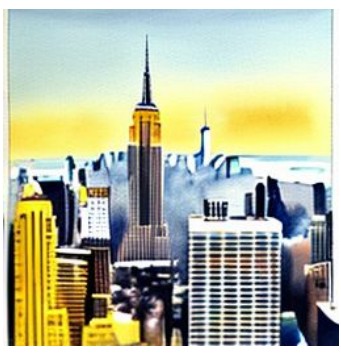 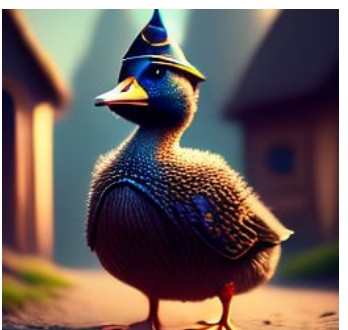 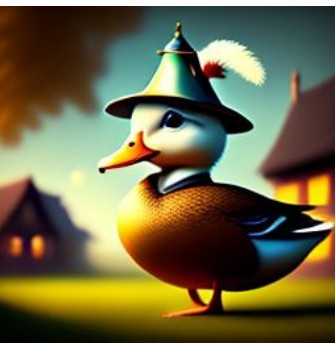

Downtown New York City at sunrise. detailed ink wash          medieval duck that lives in a village, 4k digital art

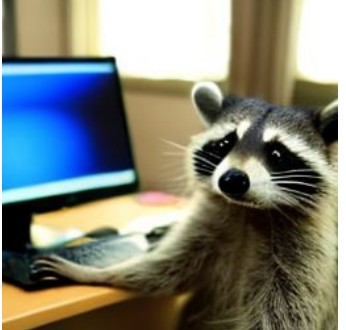 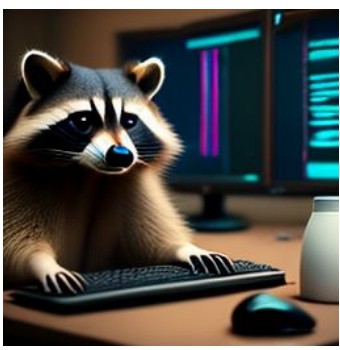 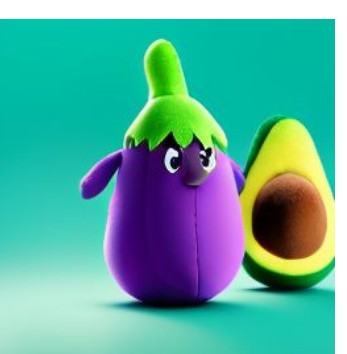 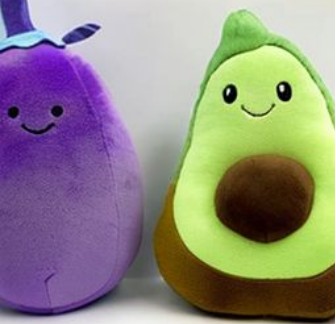

A photo of a confused racoon in computer programming class

an eggplant and an avocado, stuffed toys next to each other

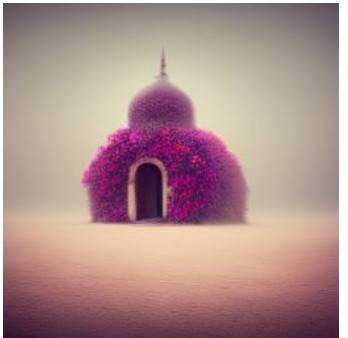 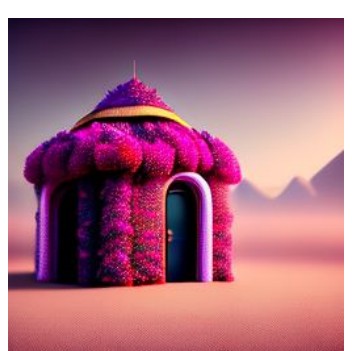 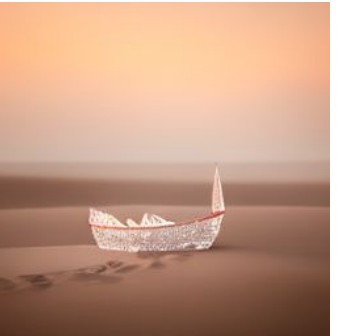 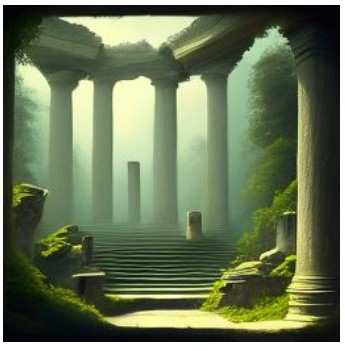

A building made up of flowers, in the middle of the empty sand, pink and purple, dreamy, foggy, photograph

A boat made up of crystals, in the middle of the empty sand, cream and orange, dreamy, foggy, photograph

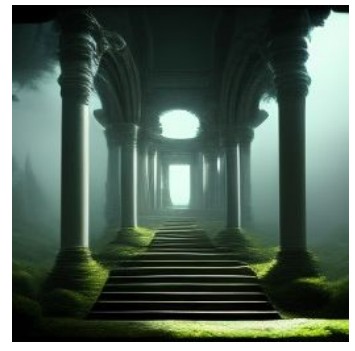 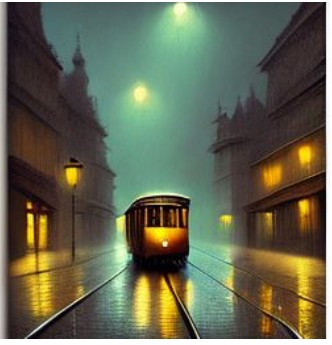

Temple in ruins, epic, forest, stairs, columns, cinematic, detailed, atmospheric, epic, concept art, matte painting, background, mist, photo-realistic, concept art, volumetric light, cinematic epic, 8k

A futuristic street train a rainy street at night in an old European city. Painting by David Friedrich, Claude Monet and John Tenniel

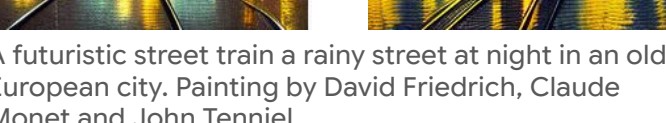

Figure A9: Additional qualitative comparisons between images generated from 3.1B and 10.5B Fluid model.

**3.1B (256x256)**  **3.1B (512x512)**  **10.5B (256x256)**

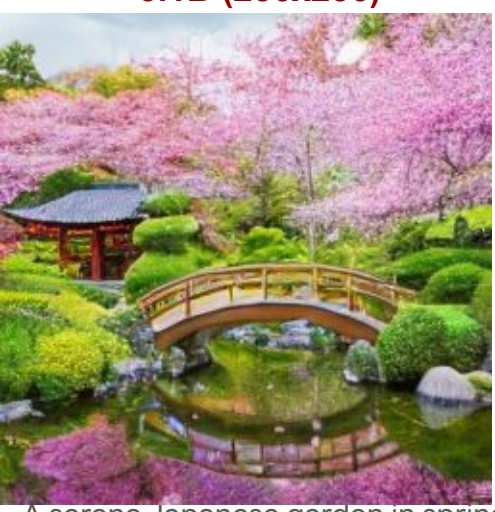 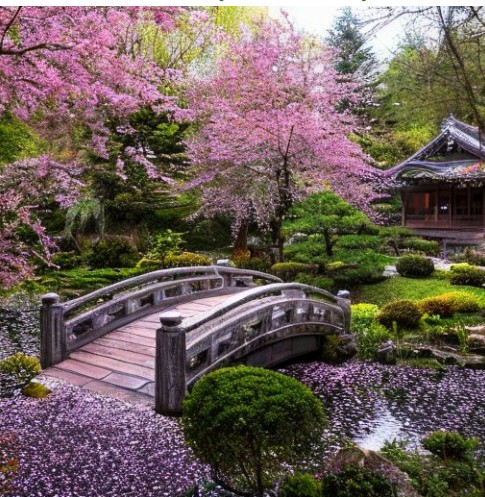 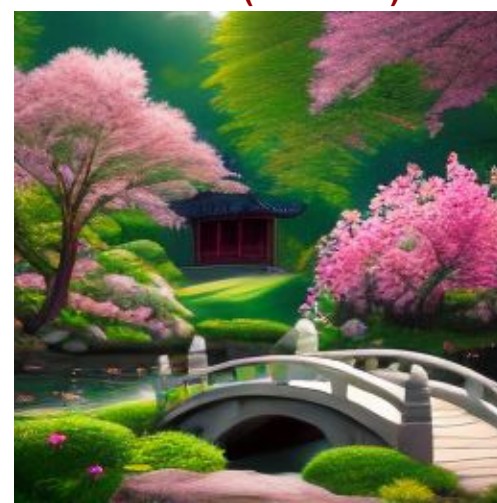

A serene Japanese garden in spring, with cherry blossom trees in full bloom, their pink petals floating gently in the breeze. A stone bridge arches over a tranquil koi pond, and a traditional tea house sits in the background, surrounded by perfectly manicured greenery.

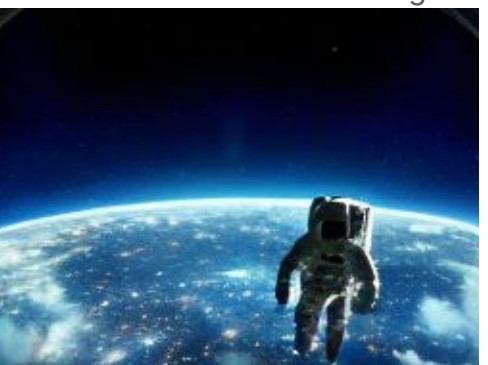 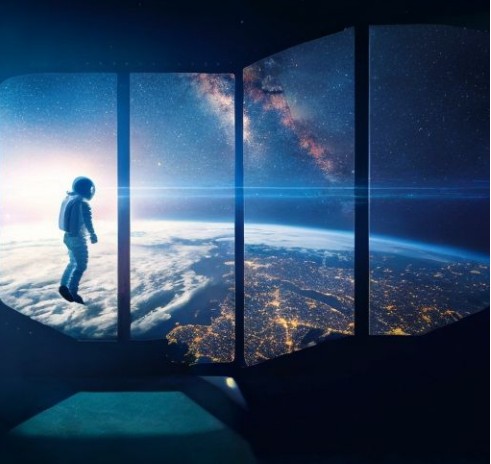 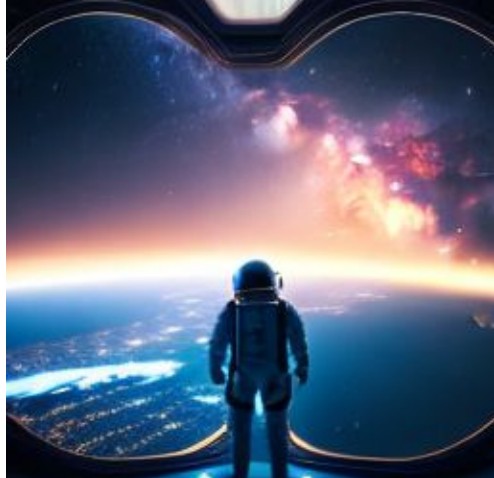

An astronaut floating weightlessly inside a space station observatory, gazing out at a panoramic view of Earth with swirling clouds, city lights, and the vast expanse of stars and galaxies beyond.

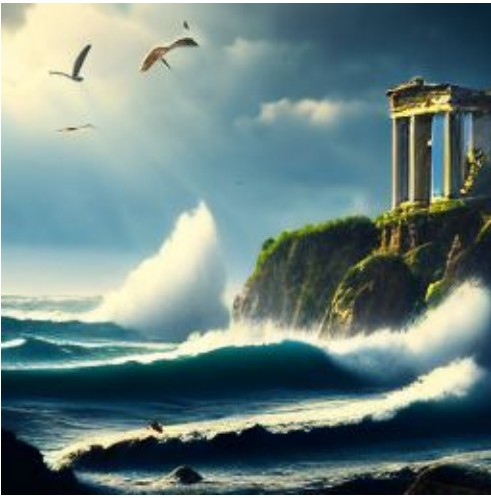 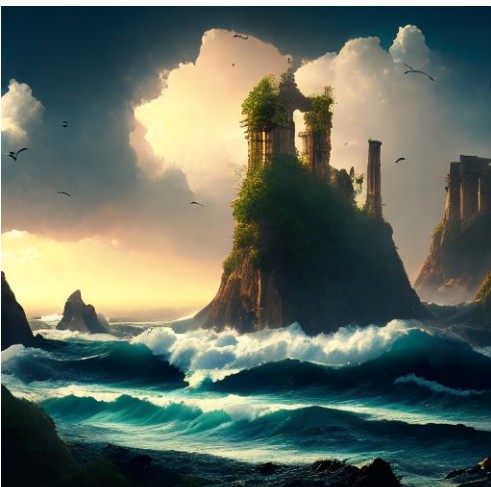 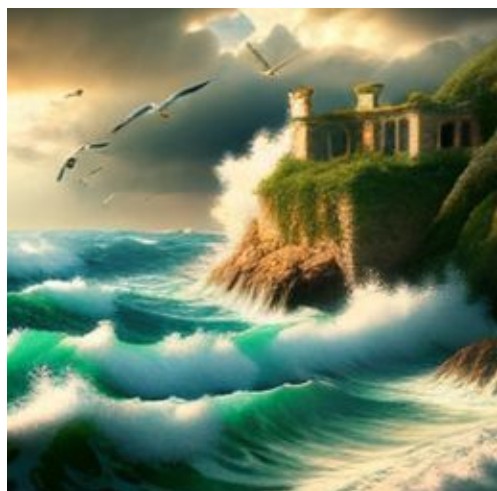

A dramatic seascape at dawn, where towering waves crash against jagged cliffs topped with ancient ruins overgrown with vines, seagulls soaring above, and the first rays of sunlight breaking through stormy clouds.

Figure A10: Qualitative comparisons between images generated from 3.1B and 10.5B Fluid model with 256x256 resolution, and 3.1B model with 512x512 resolution.

**3.1B (256x256)**     **3.1B (512x512)**     **10.5B (256x256)**

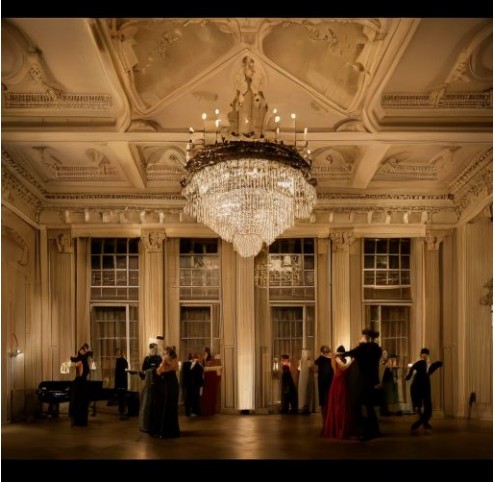
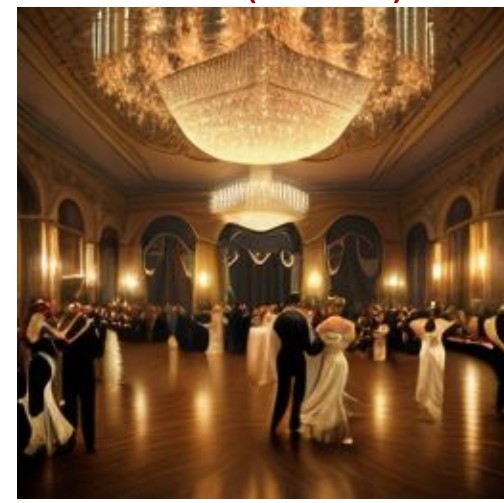

An elegant ballroom from the 1920s, filled with dancers in vintage attire, a grand chandelier casting warm light, a jazz band playing lively tunes, and Art Deco decorations adorning the walls.

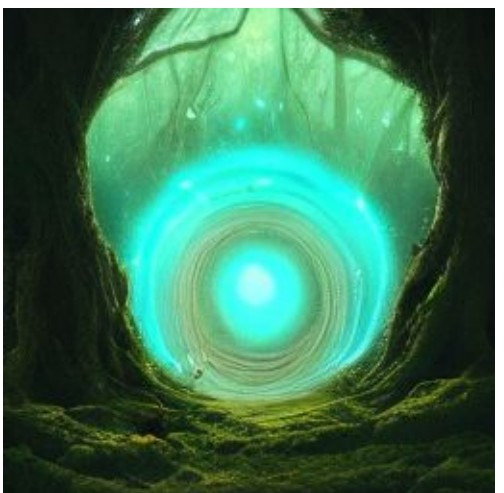
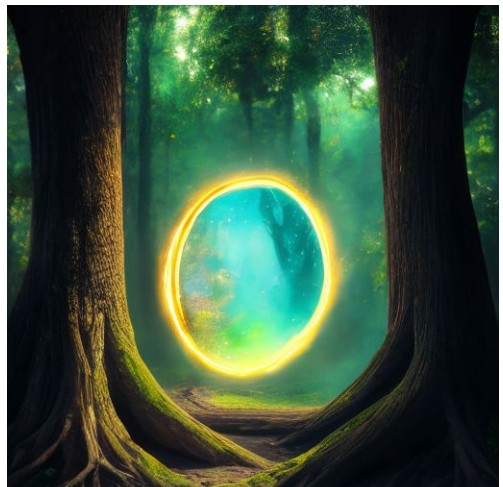
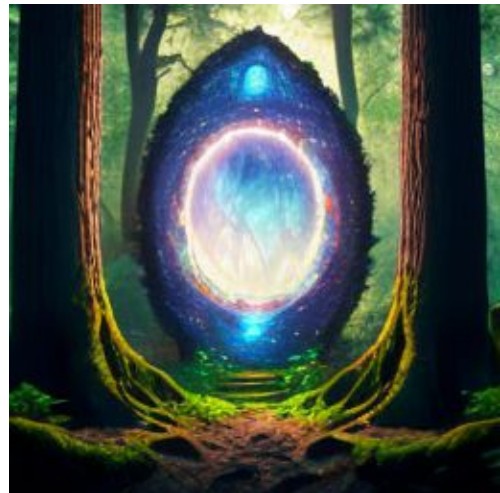

A mystical portal between two ancient trees in a forest, swirling with ethereal energy, revealing glimpses of a fantastical realm beyond.

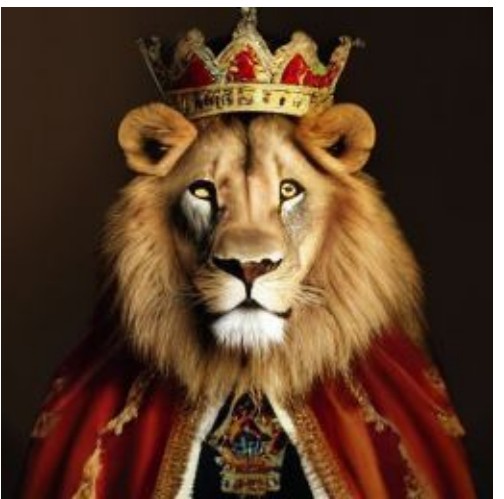
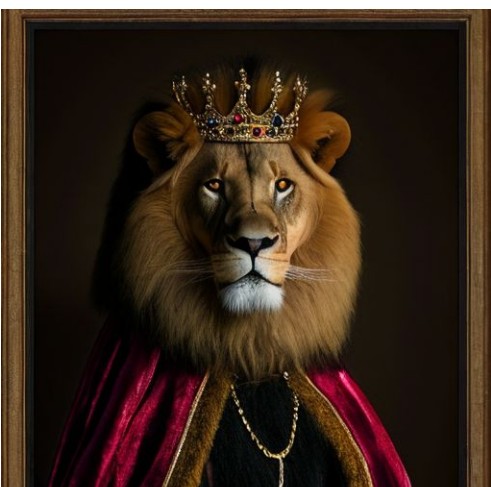
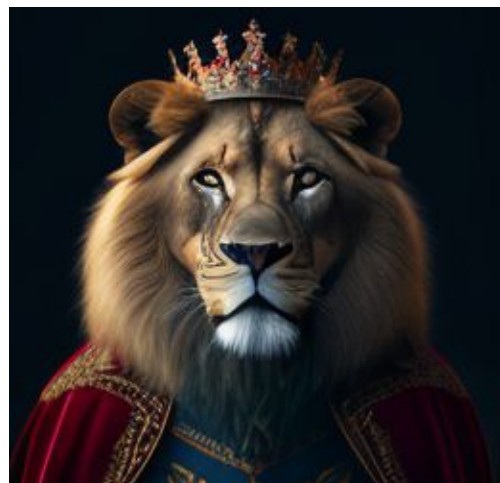

A photo of a lion wearing a regal cape and crown

Figure A11: Qualitative comparisons between images generated from 3.1B and 10.5B Fluid model with 256x256 resolution, and 3.1B model with 512x512 resolution.

**3.1B (256x256)**   **3.1B (512x512)**   **10.5B (256x256)**

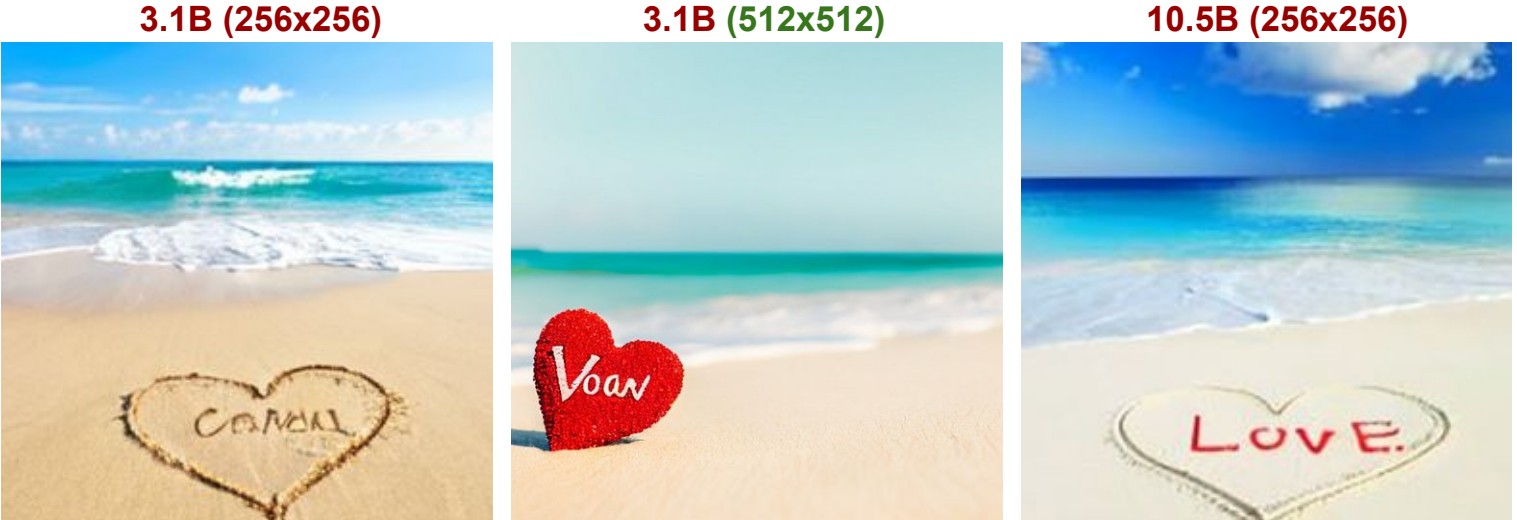

A heart with "LOVE" written in it, on the beach

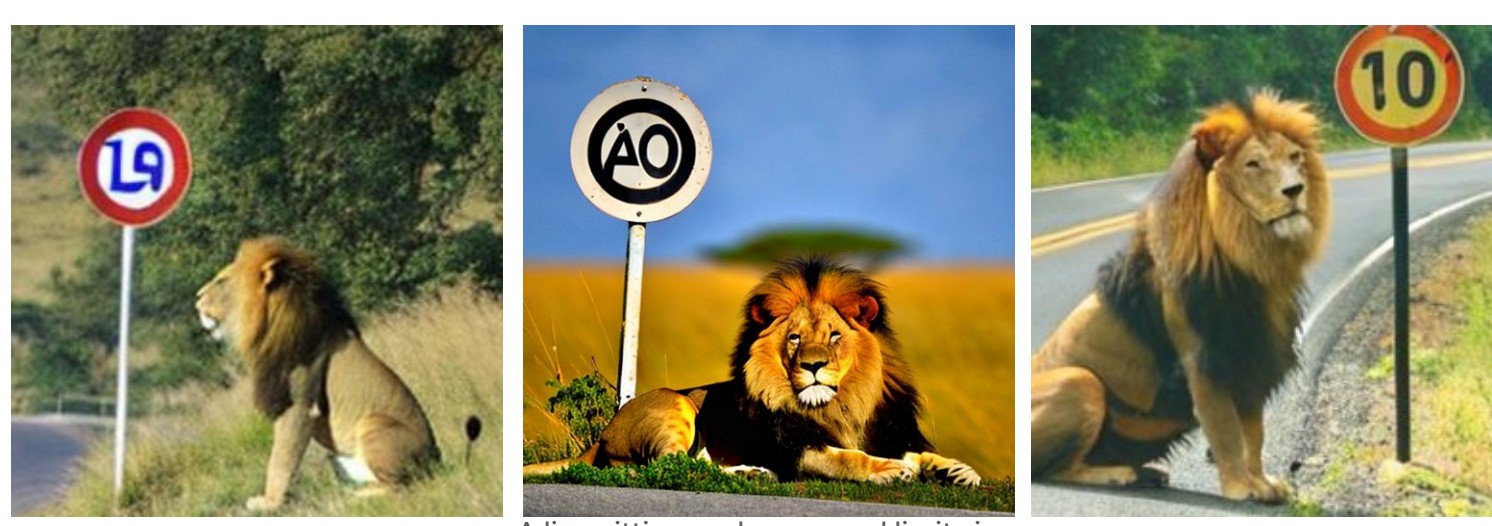

A lion sitting under a speed limit sign

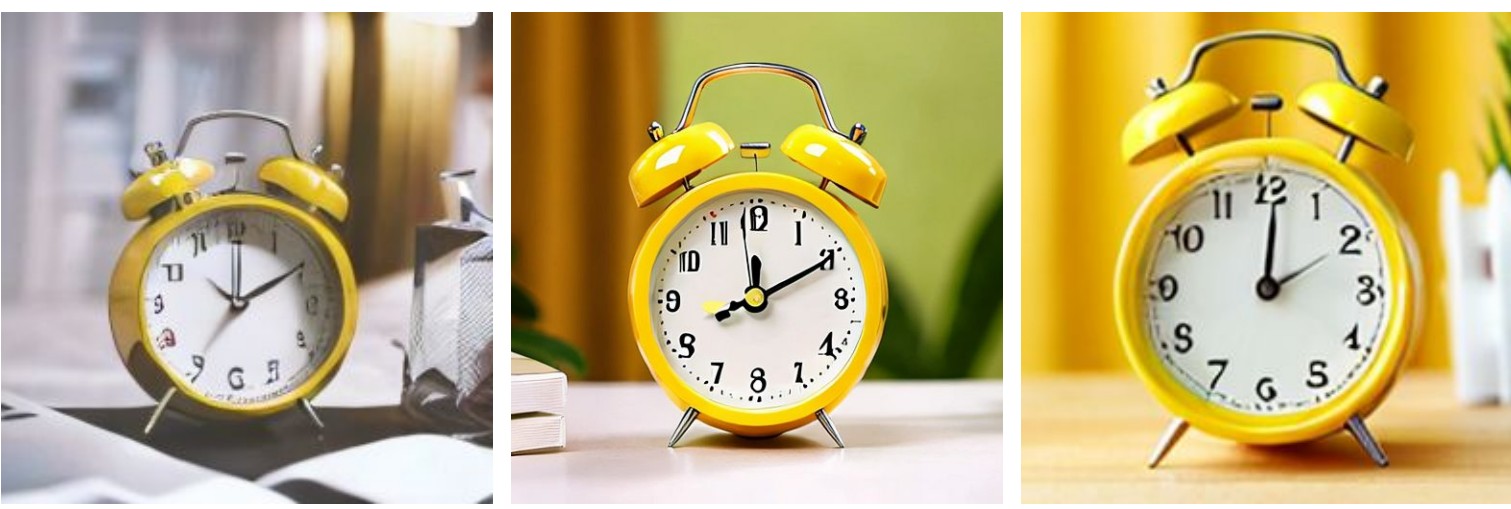

retro style yellow alarm clock with a white clock face

Figure A12: Qualitative comparisons between images generated from 3.1B and 10.5B Fluid model with 256x256 resolution, and 3.1B model with 512x512 resolution.

| **LlamaGen(256x256)** | **Fluid 3.1B** | **Fluid 3.1B (512x512)** | **Fluid 10.5B** |
|:---:|:---:|:---:|:---:|

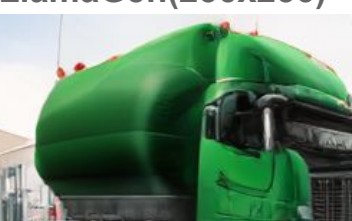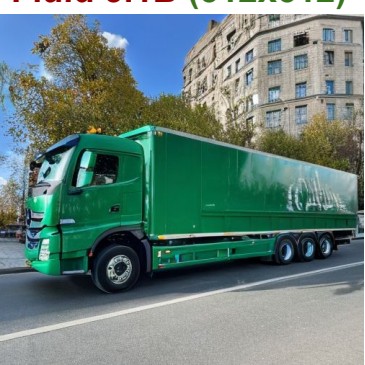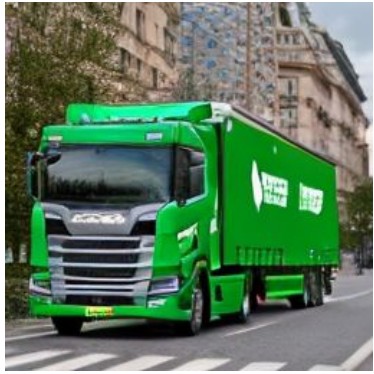

A large green truck on a city street.

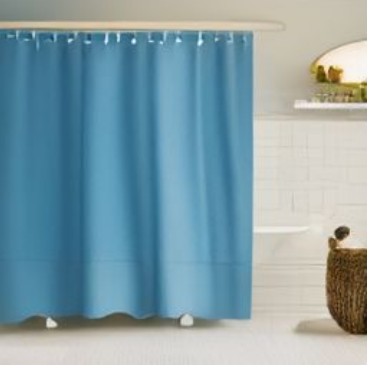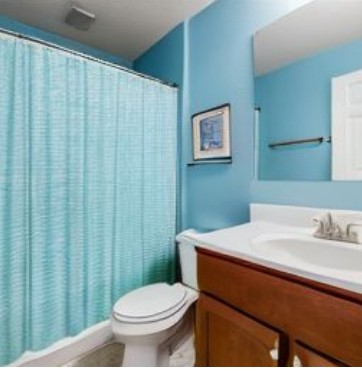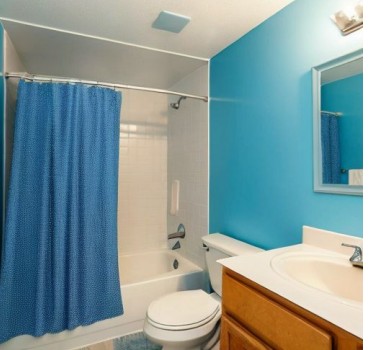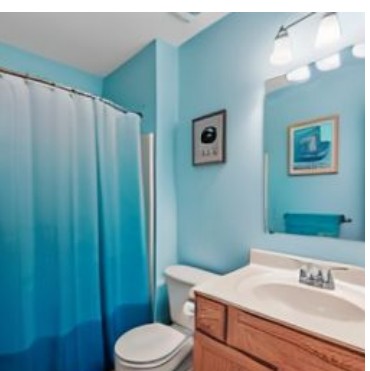

A bathroom with a blue shower curtain and blue walls.

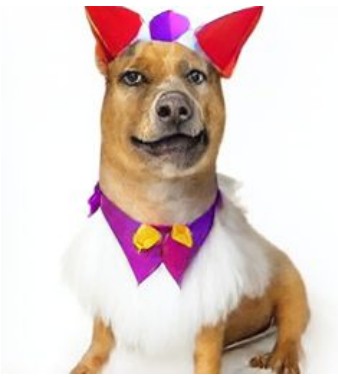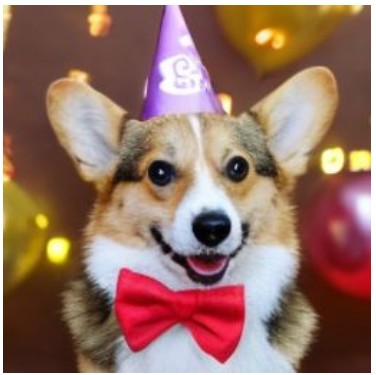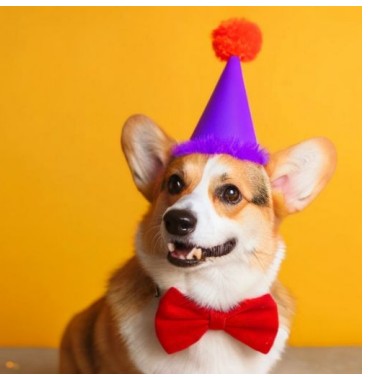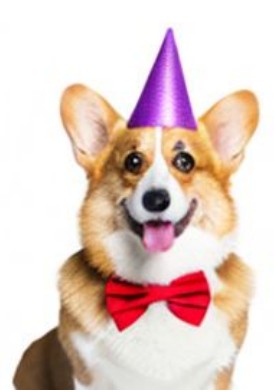

a corgi wearing a red bowtie and a purple party hat

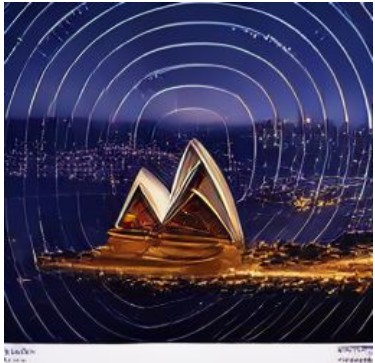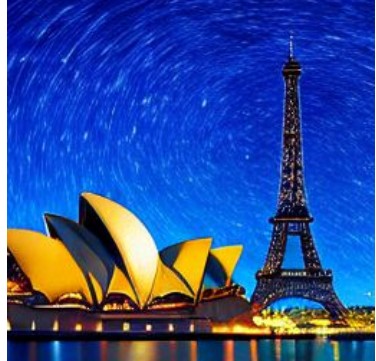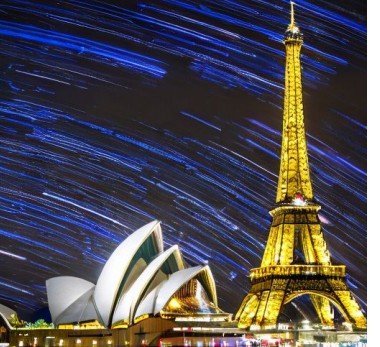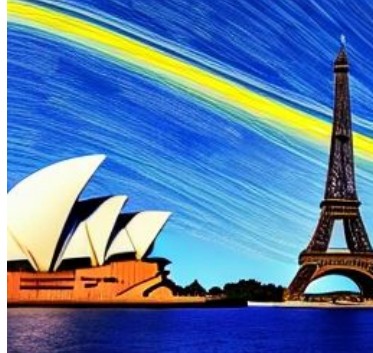

A close-up high-contrast photo of Sydney Opera House sitting next to Eiffel tower, under a blue night sky of roiling energy, exploding yellow stars, and radiating swirls of blue

Figure A13: Qualitative comparisons between images generated from 3.1B and 10.5B Fluid model and LlamaGen( Sun et al. (2024)) stage 1 model.

| LlamaGen(512x512) | Fluid 3.1B | Fluid 3.1B (512x512) | Fluid 10.5B |
|---|---|---|---|

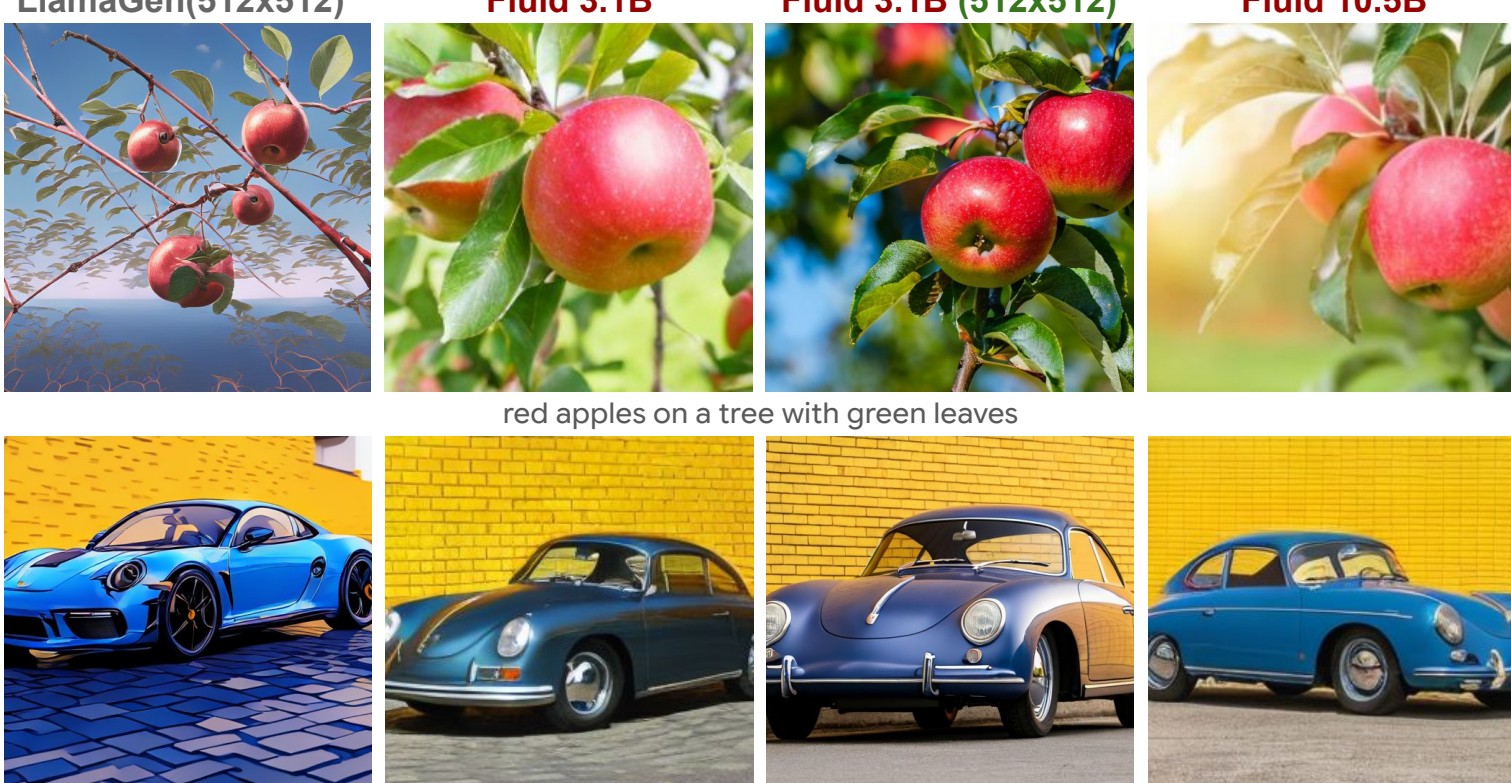

red apples on a tree with green leaves

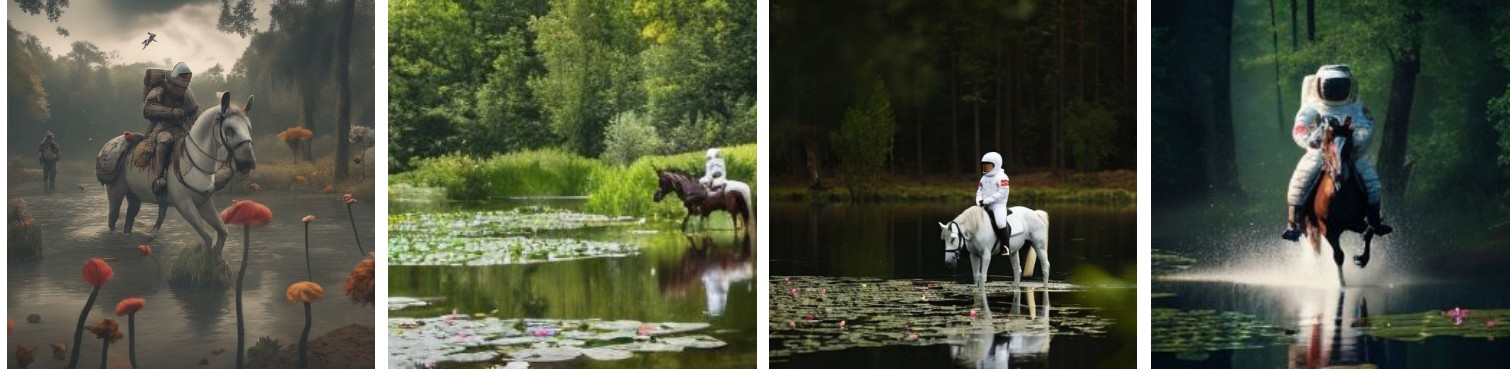

A blue Porsche 356 parked in front of a yellow brick wall

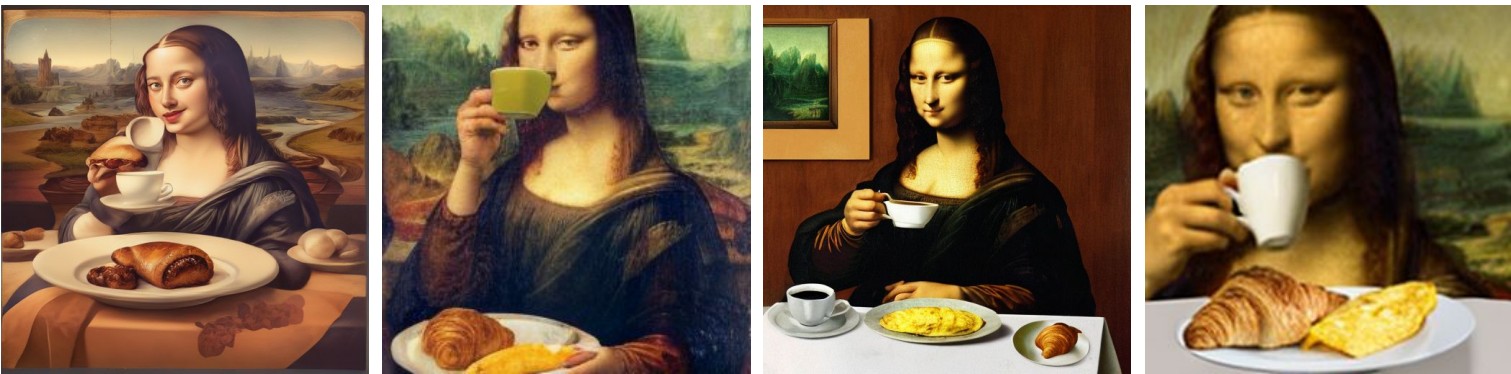

A photo of an astronaut riding a horse in the forest. There is a river in front of them with water lilies

a photograph of the mona lisa drinking coffee as she has her breakfast. her plate has an omelette and croissant

Figure A14: Qualitative comparisons between images generated from 3.1B and 10.5B Fluid model and LlamaGen( Sun et al. (2024)) stage 2 model.

image generation. In Figure A13, we compare our models with LlamaGen's stage 1 model, and in Figure A14, with its stage 2 model. The comparisons show that our Fluid models achieve significantly better alignment with textual descriptions, while LlamaGen struggles to capture certain concepts accurately. For instance, the stage 1 model generates an incorrect dog breed for "corgi" and ignores the Eiffel Tower mentioned in the text. Similarly, the stage 2 model fails to generate the correct Porsche model, a water lily, and Mona Lisa, demonstrating notable limitations in its concept understanding.

## D    FAILURE CASES.

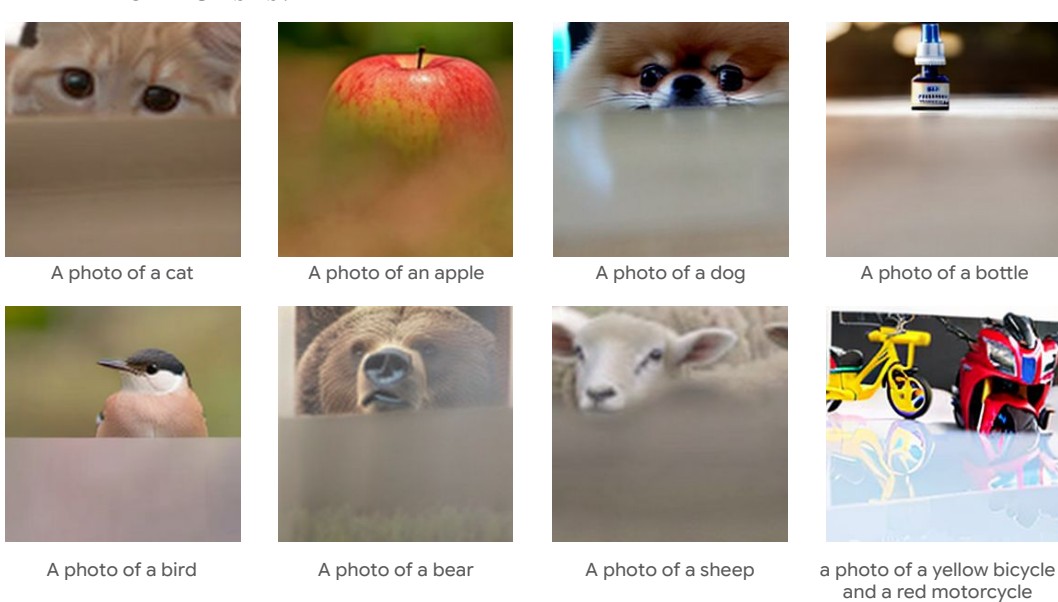

Figure A15: Failure cases for raster order generation with continuous tokens.

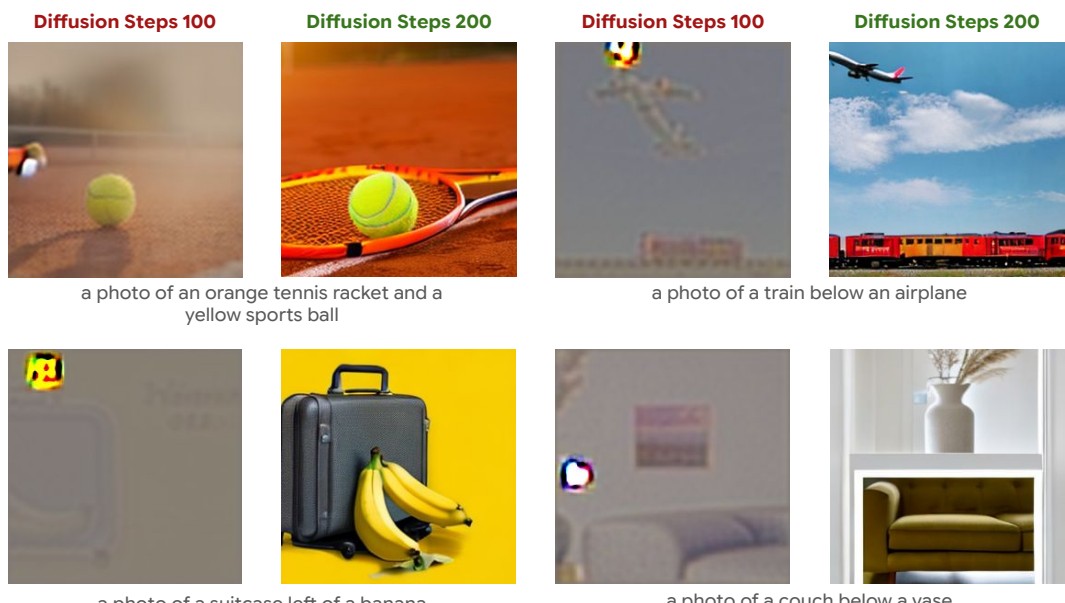

Figure A16: Failure cases for random order generation with continuous tokens. In very rare cases, an abnormal bright spots can overshadow other tokens. Increasing the diffusion steps solves this issue.

Figure A15 illustrates failure cases for raster order generation using continuous tokens. The model occasionally generates gray tokens in the lower part of the images, and once it begins, it keeps generating gray tokens and rarely recovers. This results in incomplete contents in the generated images. We suspect this issue is because the learned positional embedding for raster order generation struggles to capture the discontinuity between the end token of one line and the start token of the next. This could potentially be addressed by using 2D positional embeddings.

Figure A16 shows a failure case for random order generation with continuous tokens, *i.e.* Fluid, where the model produces abnormal bright spots. This rarely happens, and this issue can be addressed by increasing the diffusion steps, *e.g.*, from 100 to 200.