# OpenReview forum: "Fluid: Scaling Autoregressive Text-to-image Generative Models with Continuous Tokens"
_ICLR.cc/2025/Conference — ICLR 2025 Poster_

### Official Review · Reviewer_hShT · 2024-10-22

**Soundness:** 4
**Presentation:** 4
**Contribution:** 2
**Rating:** 6
**Confidence:** 4

**Summary:**

This paper proposes an auto-regressive text-to-image generation model. The authors specifically explore two aspects, namely random order v.s. raster order and continuous token v.s. discrete token. Through experiments, the authors conclude that random order with continuous tokens is better. Evaluation on MS-COCO and GenEval benchmarks indicate better performance compared with previous diffusion-based and auto-regressive-based models.

**Strengths:**

1. This paper successfully scales previous auto-regressive models based on continuous tokens up to the text-to-image setting, which is more complicated and hard-to-train than class-conditioned models.
2. The quantitative results are good given the results in Tab. 1.
3. The paper is well-written and I am able to fully follow the authors' writing.

**Weaknesses:**

1. It seems that the novelty is limited. The applied techniques are essentially the same as those proposed in [a], which can hardly bring readers new technical insights. The overall feeling of the paper is more like a technical report instead of a research paper.
2. The resolution used in the paper is 256x256, which is relatively small compared with recent models.
3. Also, I am not sure whether the comparison in Tab. 1 is fair because different systems may have been trained on different resolutions.
4. LlamaGen [b] can serve as a recently proposed baseline for comparisons.
5. Lack of qualitative comparisons and efficiency comparisons with recent models. Readers can hardly get the strengths of the proposed model against state-of-the-art ones.


[a] Autoregressive Image Generation without Vector Quantization, Li et al., 2024

[b] Autoregressive Model Beats Diffusion: Llama for Scalable Image Generation, Sun et al., 2024

**Questions:**

See weaknesses.

---

> ### Author Response · Authors · 2024-11-19
> **Response to Reviewer hShT**
>
> Thank you for your insightful comments. Besides the general response, we have also incorporated your suggestions into the revised manuscript and provided detailed explanations for each question below. We hope that our revisions and rebuttal could address your concerns and that you might consider raising the rating. Please let us know if you have any further questions or require additional results.
>
> ***Novelty***
>
> Please see the general response.
>
> ***256x256 resolution***
>
> Thank you for the observation. To address this, we trained a 3B Fluid model capable of generating 512x512 resolution images by fine-tuning the 256x256 model. Specifically, we interpolated the position embeddings and fine-tuned the model with 512x512 images for an additional 500k iterations using a constant learning rate of 1e-5. A detailed comparison of the results is provided in Section C.3 and Figure A10 and A11 in the **updated supplementary material**.
>
> Our findings show that the 512x512 images capture finer details and exhibit an overall improvement in visual quality. However, this comes with a trade-off between resolution, sequence length, and computational cost. Higher resolution requires increased sequence length, leading to greater computational demands. By balancing these factors, we demonstrate the feasibility of scaling to higher resolutions while highlighting the associated trade-offs.
>
> ***Different resolution for eval metric***
>
> Thank you for raising this concern. Regarding evaluation metrics, differences in resolution should not affect the fairness of the comparison. We adhere to the community standard for computing zero-shot FID on MSCOCO. Specifically, all generated and real images are resized to 299x299 before feature extraction using the Inception-V3 network and subsequent distance calculation. This standardization ensures that resolution discrepancies do not bias the evaluation. We will clarify this in the revised version
>
> ***LlamaGen as baseline***
>
>  Thank you for pointing out LlamaGen[1] as a potential baseline. LlamaGen (0.7B) only scores 0.32 on the GenEval benchmark, while our smallest Fluid 0.3B model achieves 0.62, and the largest 10.5B model achieves 0.69.
>
> We also used the same set of prompts provided by LlamaGen and conducted a visual comparison. The detailed results are included in Section C.4 in the **updated supplementary material**, Figure A13 and A14. Overall, our findings indicate that Fluid generates images of higher quality, with more intricate details and better text alignment, attributable to the advantages of continuous tokens.
>
> ***Qualitative comparison***
>
> Thank you for your feedback. We respectfully note that the primary objective of our work is not to build the best-performing text-to-image model, but rather to conduct the first systematic study on the scaling behavior of autoregressive vision models.
>
> We are the first to scale an autoregressive text-to-image model to the 10.5B parameter level, demonstrating consistent performance improvements as model size increases. Additionally, we identified key design principles—such as using continuous tokens and random-order generation—that enable autoregressive models to scale more effectively.
>
> While qualitative and efficiency comparisons with state-of-the-art models are valuable, we believe our findings make a significant contribution by advancing the understanding of scaling in autoregressive vision models and providing actionable insights to guide future research in this area.
>
> [1] Autoregressive Model Beats Diffusion: Llama for Scalable Image Generation, Sun et al., 2024

---

> > ### Author Response · Authors · 2024-11-23
> > **Look forward to your post-rebuttal feedback!**
> >
> > Dear Reviewer hShT,
> >
> > Thanks again for your insightful suggestions and comments. Since the deadline of discussion is approaching, we are happy to provide any additional clarification that you may need.
> >
> > In our previous response, we have carefully studied your comments and made detailed responses summarized below:
> > - Provided further explanation of our empirical novelty and contributions to the community.
> > - Conducted additional experiments to evaluate the performance of higher-resolution image generation (512x512).
> > - Performed additional experiments to demonstrate the improvement of our approach over LlamaGen, as suggested.
> > - Offered additional discussion on ensuring fairness in quantitative comparisons.
> >
> > We hope that the provided new experiments and the additional explanation on our empirical novelty have convinced you of the contributions of our submission.
> >
> > Please do not hesitate to contact us if there's additional clarification or experiments we can offer. Thanks!
> >
> > Thank you for your time!
> >
> > Best,
> > Authors

---

> > > ### Author Response · Authors · 2024-12-01
> > > **Looking forward to your response!**
> > >
> > > Dear Reviewer hShT,
> > >
> > > Thank you again for the insightful comments and suggestions! Since there are only very few days left in the discussion period, we sincerely look forward to your follow-up response.
> > >
> > > We have made significant efforts to provide additional experimental justifications and clarifications in our rebuttal. We also wanted to mention that all other reviewers have recommended acceptance of our submission. If you have any new questions or experiment requests, please feel free to let us know. We are happy to provide additional explanations and evidence to illustrate the contribution of our paper.
> > >
> > > Thank you again for your time!
> > >
> > > Best,
> > > Authors

---

> > > > ### Comment · Reviewer_hShT · 2024-12-02
> > > >
> > > > Thanks for the detailed reply and for conducting new experiments. It addresses my concerns partially. For the resolution used for comparison, I respectfully disagree with the authors that different resolutions should not affect fairness. After all, the training difficulties of low resolution and high resolution are dramatically different. A model/architecture that performs well on low-resolution images can also achieve equivalently good performance at high resolutions, especially compared with the state-of-the-art models that have been trained on high-resolution images.
> > > >
> > > > If the authors can show that the newly trained 512 model can also achieve performance as well as that shown in Tab, I am willing to increase my score.

---

> > > > > ### Author Response · Authors · 2024-12-02
> > > > > **Additional results to address reviewer's new question**
> > > > >
> > > > > Dear Reviewer hShT,
> > > > >
> > > > > Thanks for your reply! We apologize for any confusion and thank you for your clarification. To ensure a fair comparison between Fluid and recent models trained at 512×512 resolution, we evaluate the FID and GenEval scores of the newly trained Fluid 3.1B (512x512) model on 512x512 resolution. This model achieves an FID of 6.6 and a GenEval score of 0.68, which is on par with the performance at 256x256 resolution. The visual quality of this high-resolution model is also much better than the Fluid 3.1B (256x256) model, as shown in Figures A10 and A11 in the updated supplementary material. We will include these results in the final paper. We hope that this result further strengthens the performance of Fluid across different resolutions and addresses your concerns about fair comparison.
> > > > >
> > > > > Please do not hesitate to contact us if you need any further clarification or experiment results. Thank you!
> > > > >
> > > > > Best wishes,
> > > > >
> > > > > Authors

---

> > > > > > ### Comment · Reviewer_hShT · 2024-12-02
> > > > > >
> > > > > > Thanks for the new results. With the new experiments, I think the strengths of the paper outweigh the weaknesses slightly. Therefore, I increase my score to 6.

---

> > > > > > > ### Author Response · Authors · 2024-12-02
> > > > > > > **Thank you!**
> > > > > > >
> > > > > > > We greatly appreciate your positive feedback! We are happy to hear that your concerns are resolved.
> > > > > > >
> > > > > > > Thanks again for your valuable time and your helpful constructive feedback!
> > > > > > > Please do not hesitate to let us know If there are any further clarifications or experiments we can offer.
> > > > > > >
> > > > > > > Best wishes,
> > > > > > >
> > > > > > > Authors

---

### Official Review · Reviewer_QGyS · 2024-10-28

**Soundness:** 3
**Presentation:** 3
**Contribution:** 2
**Rating:** 6
**Confidence:** 3

**Summary:**

This work investigates the performance of autoregressive models in generating high-quality images from text prompts, focusing on the use of continuous versus discrete tokens and the impact of token generation order. The authors find that models using continuous tokens produce superior visual quality compared to those with discrete tokens, indicating that continuous representations capture more nuanced information. Additionally, they reveal that random-order generation outperforms fixed raster-order generation, highlighting the significance of the token generation sequence on image quality. Employing various evaluation metrics, the study compares its results with leading diffusion models and autoregressive models, reporting favorable outcomes. Overall, the research provides empirical insights into the scaling behavior of text-to-image generation models, emphasizing the importance of token representation and generation strategy in enhancing autoregressive models' capabilities for high-resolution image synthesis.

**Strengths:**

1. The writing in the article is clear and easy to understand.

2. The improvements to the original autoregressive model, such as continuous token and random generation order, are both insightful and empirically effective techniques. These empirical results provide crucial experimental evidence that supports some of the fundamental beliefs about autoregressive image generation.

**Weaknesses:**

1. Although this article is the first to compare the roles of token representation and generation order together in the autoregressive image generation, the techniques it examines are not novel or originally proposed by the authors. For instance, line 189 notes that the continuous token method is derived from a prior work, and line 287 mentions that the implementation of random order also originates from an earlier study. This work does not optimize or update the technical details of these methods. I highly appreciate the article’s empirical study on autoregressive image generation, but the scope seems limited to experimental evaluation of existing techniques without introducing new methods or innovations.

2. Autoregressive image generation still lacks a standardized technical framework, and both its training and inference processes remain complex and non-standardized. The approach involves various technical details beyond just token representation and generation order. I highly appreciate the authors' efforts to highlight these two factors amidst the complexity. However, different token representations and generation orders introduce other technical variations, making it challenging to ensure absolute fairness in comparisons. This limitation may prevent us from making strictly definitive claims based on the comparative results.

**Questions:**

1. In the experimental results presented in Figure 6, it seems that with the minimal number of parameters, continuous tokens do not outperform discrete tokens. Would the authors be able to provide further analysis on why this might be the case?

2. In the implementation of continuous tokens, an extra diffusion head is used, whereas no comparable diffusion structure is applied for discrete tokens. If possible, could the authors conduct further ablation studies to examine the impact of this diffusion head?

3. The authors also mention that these techniques vary not only in performance but in efficiency as well. Would it be possible to provide a more detailed comparison of inference efficiency?

---

> ### Author Response · Authors · 2024-11-19
> **Response to Reviewer QGyS**
>
> Thank you for your insightful comments. Besides the general response, we have also incorporated your suggestions into the revised manuscript and provided detailed explanations for each question below. We hope that our revisions and rebuttal could address your concerns and that you might consider raising the rating. Please let us know if you have any further questions or require additional results.
>
> ***Novelty***
>
> Please see the general response for a detailed response. We believe novelty not only comes from the technical side, e.g., proposing a new model, but also comes from interesting findings that can inspire the future direction. The latter can sometimes be more important.
>
> ***Comparison Fairness***
>
> Thank you for raising this concern. We took careful steps to ensure that our comparisons are both plausible and fair, despite the inherent complexity of autoregressive image generation. Specifically:
>
> 1. Token Representations:
>    - For discrete tokens, we used state-of-the-art VQ tokenizers, as employed in MUSE (but we re-trained it with larger datasets and improved recipe).
>    - For continuous tokens, we adopted the Stable Diffusion tokenizer, which we found through ablations to produce the best FID scores.
>
> 2. Inference Hyperparameters:
>     We conducted extensive sweeps over inference hyperparameters (e.g., temperature and classifier-free guidance) for all configurations, reporting the best results for each model to ensure fairness.
>
> 3. Training Hyperparameters:
>    - All training hyperparameters, such as dataset, learning rate, and number of iterations, were kept consistent across configurations.
>    - The only exception was the learning rate schedule: we used a cosine schedule for MUSE as it led to better results compared to a constant schedule.
>
> While we acknowledge that achieving absolute fairness in such comparisons is inherently challenging, these measures were taken to minimize confounding factors as much as possible. We believe this approach provides a robust foundation for evaluating the roles of token representation and generation order in scaling autoregressive models.
>
> ***Q1 Discrete tokens are better in small models***
>
> Thank you for pointing this out. This observation aligns with our expectations. The categorical distribution used in discrete tokens is inherently simpler and easier to learn compared to the continuous distribution. While this simplicity allows smaller models, which have limited capacity, to perform better with discrete tokens, it comes at the cost of being lossy and less capable of modeling complex distributions.
>
> In contrast, continuous tokens offer greater expressive power and flexibility, but their complexity makes it harder for smaller models to optimize effectively. This trade-off highlights why discrete tokens are better suited for smaller models, while continuous tokens exhibit their advantages as model capacity increases.
>
> ***Ablation on diffusion head***
>
> Thank you for raising this point. The following Table S1 presents a pilot study on the effect of the diffusion head for continuous tokens. We trained the smaller model with 277M parameters using random order continuous tokens with a T5-XL text encoder for 500k steps. The FID score shows consistent improvements as we enlarge the depth/width of the diffusion head. We choose to use 6 layers with the same width as the backbone transformer in our Fluid models to balance performance and efficiency.
>
> <Table S1. FID vs. different diffusion head sizes.>
>
> |Depth|Width|FID|
> |-|-|-|
> |3|512|10.46|
> |3|1024|9.95|
> |6|1024|9.38|
>
> Incorporating a diffusion head for discrete tokens is inherently challenging because the diffusion process is designed to model continuous distributions. As a result, applying this structure to discrete tokens would require significant modifications, and it would diverge from the current architecture's principles.
>
> ***Q3 Inference efficiency comparison***
>
> We provide a detailed comparison of inference speeds across different configurations in Table S2. The key details of the setup are as follows:
>
> Random-Order Generation:
> - Images are generated in 64 steps.
> - The inference speed is measured on a single H100 PCIe GPU, using the largest possible batch size (128).
>
> Raster-Order Generation:
> - Images are generated with KV-cache enabled.
> - The inference speed is also measured on a single H100 PCIe GPU, with the largest batch size supported (64). The largest batch size is smaller than the random-order models because KV-cache takes more GPU memory.
> - The continuous token + raster order setting is particularly slow because the DiffLoss module can only use a batch size of 64 at each autoregressive step, which largely wastes the capacity of the GPU.
>
> <Table S2. Inference speed comparison.>
>
> |Token|Order|Speed (s/img)|
> |:-|-|-|
> |Continuous|Random|0.59517|
> |Continuous|Raster|1.01638|
> |Discrete|Random|0.5035|
> |Discrete|Raster|0.1781|

---

> ### Author Response · Authors · 2024-11-23
> **Look forward to your post-rebuttal feedback!**
>
> Dear Reviewer QGyS,
>
> Thanks again for your insightful suggestions and comments. Since the deadline of discussion is approaching, we are happy to provide any additional clarification that you may need.
>
> In our previous response, we have carefully studied your comments and made detailed responses summarized below:
> - Provided further explanation of our empirical novelty and contributions to the community.
> - Offered additional discussion on ensuring fairness in quantitative comparisons.
> - Conducted new experiments to study the effect of diffusion head sizes.
> - Compared inference efficiency across different configurations through additional experiments.
>
> We hope that the provided new experiments and the additional explanation on our empirical novelty have convinced you of the contributions of our submission.
>
> Please do not hesitate to contact us if there's additional clarification or experiments we can offer. Thanks!
>
> Thank you for your time!
>
> Best,
> Authors

---

### Official Review · Reviewer_y1Ue · 2024-11-03

**Soundness:** 3
**Presentation:** 4
**Contribution:** 3
**Rating:** 5
**Confidence:** 5

**Summary:**

This paper investigates the scaling behaviors of autoregressive models in text-to-image generation, focusing on key factors: token type (discrete vs. continuous) and generation order (random vs. raster). Experiment results demonstrate a power-law relationship between the validation loss and model sizes, and that continuous tokens with random-order generation yield the best performance on evaluation metrics. Leveraging these findings, the authors introduce Fluid, a 10.5B random-order autoregressive model with continuous tokens, outperforming previous models on benchmarks such as FID and GenEval.

**Strengths:**

- Achieve SOTA performance with zero-shot FID of 6.16 on MS-COCO 30K, and an overall score of 0.69 on the GenEval benchmark.
- The paper is well-organized and easy to follow, with robust training data, strategies, and solid experimental setups.
- Extensive experiments reveal clear relations between the four configurations of token types and generation order.

**Weaknesses:**

- This paper ablated two types of existing technologies, with their scaling behaviors. The experiments are well-designed, and the authors' observations appear reasonable. However, some additional insights are needed.
- I appreciate the authors' efforts in sharing extensive experimental results, but providing an actionable conclusion would add more value than observations alone. If the authors could provide insights or equations that can guide future training of large-scale text-to-image models—such as identifying optimal settings when constrained by FLOPs, image samples, or model size—it would significantly benefit the field.

**Questions:**

- My questions arise naturally from the "Weaknesses", I believe the authors may have more insights beyond the experiments results provided.
  - What underlying factors drive the observed correlation between metrics and the linear relationship between validation loss and the metrics?
  - What should be considered the most "human-aligned" evaluation criterion for future research?
  - More analysis of the benefits of the continuous tokens and random-order prediction should be explained, especially during the scaling up, e.g., why does random-order prediction yield a larger loss but better visual quality?
- Similar to the "Scaling Law" curves in the language model community, an ablation on the number of training examples across the four configurations could be beneficial.
- Is there any "Emergence" phenomena during training, such as a sudden increase in metric scores?
- I would consider raising my scores if these points are addressed.

---

> ### Author Response · Authors · 2024-11-19
> **Response to Reviewer y1Ue (1/2)**
>
> Thank you for your insightful comments. Besides the general response, we have also incorporated your suggestions into the revised manuscript and provided detailed explanations for each question below. We hope that our revisions and rebuttal could address your concerns and that you might consider raising the rating. Please let us know if you have any further questions or require additional results.
>
> ***Q1-1 Validation loss & metric***
>
> *Correlation Between Validation Loss and Metrics:*
>
> Validation loss directly reflects the training target, while metrics like FID and GenEval capture different aspects of generation quality and alignment. In NLP, this gap is smaller because validation metrics (e.g., next-token prediction) are the same as the training objective. In contrast, for autoregressive vision models, the discrepancy is larger because:
> -   All autoregressive steps occur in the latent space, and validation loss is measured there.
> -   The final image is decoded from the latent space and then evaluated using metrics, introducing additional complexity.
>
> This separation between latent-space training and pixel-space evaluation may drive the observed differences, and we highlight this as a key finding in our study.
>
> *A linear relationship between validation loss and metrics:*
>
> They are correlated because they all measure the generation capability (but from different perspectives). The relationship is highly linear under the regime we evaluated, which suggests the model is highly scalable (at least to the extent we explored). As pointed out in line 485, such a linear relationship will be broken at some point.
>
> ***Q1-2 Human-aligned evaluation criterion***
>
> Identifying a universally human-aligned evaluation metric is a broader challenge not specific to this paper. While benchmarks like GenEval are promising, they are limited in scope. A potential direction for future work is creating more complex prompts and datasets to evaluate nuanced relationships and subject matter. Such efforts could pave the way for better automatic evaluation methods aligned with human perception.
>
> ***Q1-3 Why random order is better with higher loss***
>
> Thank you for your thoughtful question. Random-order generation introduces additional complexity to the image generation task by using a masked version of the original image as input. This makes the model's task inherently harder, as it must predict tokens in arbitrary orders without relying on positional continuity. While this results in a larger training loss, it encourages the model to learn more robust representations for the image-text alignment.
>
> During scaling, this robustness translates into better generalization and higher visual quality, as the model can generate coherent and detailed images even in the presence of challenging input scenarios. Additionally, random-order generation reduces the positional bias that sequential generation might introduce, leading to improved flexibility and accuracy in token prediction.
>
> The confusion may come from directly comparing the loss of random order with the loss of raster order. We emphasize that such validation loss is not directly comparable across different training setups. A simple demonstration is that we can train the 1.1B model with continuous tokens and random order for 500k iterations, using different mask ratios. As the masking ratio increases, the loss gets consistently higher but the FID first increases then decreases.
>
> <Table S1. Validation loss and FID with different masking ratios.>
>
> | Masking Ratio   |   0.4 |   0.6 |   0.8 |   0.9 |
> |-----------------|------:|------:|------:|------:|
> | Validation Loss | 0.299 | 0.303 | 0.318 | 0.335 |
> | FID             |  6.93 |  6.74 |  7.08 |  8.15 |

---

> ### Author Response · Authors · 2024-11-19
> **Response to Reviewer y1Ue (2/2)**
>
> ***Q2 Ablation on number of training samples***
>
> Thank you for highlighting this important aspect of scaling. To address this, we trained a 1.1B Fluid model with different amounts of image-text pairs for 500k iterations, and summarized the results in the table below.
>
> <Table S2. Scaling behavior of Fluid 1.1B model on different data sizes>
>
> | Dataset Portion (%) |   0.05 |    0.1 |   0.25 |    0.5 |      1 |      5 |     10 |    100 |
> |:---------------------|-------:|-------:|-------:|-------:|-------:|-------:|-------:|-------:|
> | FID                 |  52.70 |  12.33 |   7.72 |   7.23 |   6.74 |   6.70 |   6.67 |   6.73 |
>
> Our findings show that FID improves consistently as the dataset size increases, up to approximately 5-10% of the total training samples. Beyond this point, the improvements on FID plateau, likely because larger datasets require more training iterations and larger models to showcase its benefits.
>
> ***Q3 Emergence phenomena***
>
> Thank you for raising this interesting point. We have added additional qualitative comparisons between the images generated by the 3.1B model and the 10.5B model in Section C.2 in the **updated supplementary material**, Figure A8 and A9. From these comparisons, we observe an interesting phenomenon in visual quality: the 10.5B model demonstrates the ability to accurately capture and represent numbers and characters, such as correctly generating a “stop” sign with legible text. In contrast, the 3.1B model lacks this capability, failing to generate plausible or coherent characters. However, this ability is not reflected in the FID and GenEval evaluation metrics, as they do not explicitly measure it.
> Nevertheless, this suggests that as the model scales, it develops a deeper understanding of complex patterns, such as textual and numerical information, which were not evident in smaller models. These emergent abilities highlight the advantages of scaling autoregressive models for text-to-image generation.

---

> > ### Comment · Reviewer_y1Ue · 2024-11-26
> >
> > The authors' rebuttal addressed most of my concerns, and I would like to raise the score. However, I believe the paper would benefit from providing not only positive signals or insights into the effects of scaling but also practical guidance for the research community, e.g., scaling laws, improved evaluation methods, or similar frameworks could serve as valuable directions for future experiments.
> >
> > The advantages and limitations of the "mask" objective are well understood. A less constrained prior could potentially yield better results for larger-scale models. Also, further exploration of the "text-on-image" generation would be intriguing.

---

> > > ### Author Response · Authors · 2024-12-02
> > > **Discussion period ending soon, looking forward to response from Reviewer y1Ue**
> > >
> > > Dear Reviewer y1Ue,
> > >
> > > Thank you once again for your positive feedback and for expressing your willingness to raise the score!
> > >
> > > With less than one day left for the discussion period, we wanted to ensure you had a chance to review our latest response. We would be happy to provide any further clarifications or experiments you would like.
> > >
> > > Additionally, we wanted to mention that all other reviewers have recommended acceptance of our submission, and we would deeply appreciate it if you could adjust the score to reflect that our rebuttal has addressed your concerns.
> > >
> > > Thank you again for your time and insightful comments.
> > >
> > > Best wishes,
> > >
> > > Authors

---

> ### Author Response · Authors · 2024-11-23
> **Look forward to your post-rebuttal feedback!**
>
> Dear Reviewer y1Ue,
>
> Thanks again for your insightful suggestions and comments. Since the deadline of discussion is approaching, we are happy to provide any additional clarification that you may need.
>
> In the original review, you expressed the willingness to turn to the positive side if additional insights on our results and experiments examining scaling behavior with respect to data sizes were provided. We hope that the new experiments and the additional explanation on validation loss and our empirical contribution have convinced you of the contributions of our submission.
>
> A **key takeaway** from our work is continuous tokens with random order scale significantly better than other configurations and work best for large autoregressive text-to-image models. Based on this, we recommend adopting continuous tokens with random order for large models, and we believe these findings offer valuable empirical guidance for future research in autoregressive vision-language and video generative models.
>
> Please do not hesitate to contact us if there's additional clarification or experiments we can offer. Thanks!
>
> Thank you for your time!
>
> Best,
> Authors

---

> ### Author Response · Authors · 2024-11-27
> **Thank you!**
>
> Thank you for your positive feedback. We are glad to hear that we have addressed your concerns. In addition to the positive aspects, we have provided failure cases and analysis of different design choices  (please see Section D in the supplementary material). In Figure 5, we demonstrate that the validation loss follows a clear and predictable scaling law w.r.t. the number of parameters, which can potentially serve as a practical guidance for training large-scale generative models. However, we also notice that the transfer from validation loss to evaluation metrics remains unclear. The exact scaling law on evaluation metrics is a very interesting and important topic, and we leave it for future work to explore.
>
> As you pointed out, for large-scale models, a less constrained prior could potentially yield better results. We believe this is likely the reason why both continuous tokens and random order perform better, as they represent "harder" tasks and are less constrained.
>
> We have also included more "text-on-image" generation results, which are provided in **Figure A12** in the updated supplementary material.
>
> We truly appreciate that you agree to raise your score. Please let us know if anything remains unclear or if you would like to see additional results or explorations.

---

> > ### Author Response · Authors · 2024-12-01
> > **Looking forward to your response!**
> >
> > Dear Reviewer y1Ue,
> >
> > Thank you again for discussing with us!
> > In your previous response, you mentioned that *"the authors' rebuttal addressed most of your concerns"* and that you *"would like to raise the score."*
> >
> > We appreciate your feedback and noticed that the current score remains unchanged. Since there are only very few days left in the discussion period, we would love to hear whether we can offer you any additional clarifications or experiments that you find interesting, or if you have any further suggestions to help us strengthen this work even more. We sincerely look forward to your follow-up response and a reflection on the score.
> >
> > Thanks again for the effort and time you have dedicated to our work and discussion! Please don't hesitate to let us know if you have any further questions.
> >
> > Best Wishes,
> >
> > Authors

---

### Official Review · Reviewer_UT5L · 2024-11-04

**Soundness:** 3
**Presentation:** 3
**Contribution:** 3
**Rating:** 6
**Confidence:** 4

**Summary:**

The paper explores scaling autoregressive models for text-to-image generation, focusing on two key factors: discrete vs. continuous token representations, and token generation order (raster vs. random). The results show that using continuous tokens and generating in random order lead to better image quality, especially at scale. The new Fluid model, which adopts these strategies, achieves state-of-the-art results on benchmarks like MS-COCO and GenEval.

**Strengths:**

Solid Scaling Analysis: The paper does a thorough job of laying out the scaling trends. The consistent improvements with random order and continuous tokens give a clear picture of what’s working.

Concrete study on Toke type and Scan order: The results convincingly show that generating tokens in a random order outperforms the traditional raster order, especially for tasks needing a good global structure. Additionally, switching to continuous tokens noticeably enhances image quality.

This is a solid empirical paper that offers important insights into scaling autoregressive models, mostly on scaling the MAR model (Li et al., 2024). The work holds relevance for both academia and industry, with the potential for meaningful social impact.

**Weaknesses:**

I value the contribution of the paper. While I am generally positive, we can still find some imperfect points in the paper.

Methodology Largely Follows MAR (Li et al., 2024): The approach in this paper closely follows the method used in MAR, with some modifications to incorporate text condition. While these changes are valuable, the paper doesn’t go far beyond what MAR already established. Also, the discussion of scan order and token type is also a repeat of what has been done in the MAR paper.

Over-Reliance on Empirical Results: While the paper does a great job presenting empirical evidence, there’s a lack of theoretical explanation for why continuous tokens outperform discrete ones. The benefits are demonstrated, but there isn't much discussion on how continuous representations help in practice—whether it’s due to better gradient flow, more expressive capacity, or something else entirely. More in-depth analysis could add substance to the findings.

Scaling Analysis Could Be More Comprehensive: The paper primarily showcases large model almost for sure improve the performance. However, other important factors in scaling, such as the effect of dataset size, are not thoroughly explored. For instance, language model research, like Kaplan et al. (2020), investigates how increasing dataset size influences validation loss, providing insights into the relationship between data and model scaling. In contrast, all the experiments in this paper are conducted on the same subset of the WebLI dataset, leaving the impact of different data sizes unexamined.

**Questions:**

1.	Is there any specific differences compared to MAR paper?
2.	Any thoughts on why random order generation works better than raster? Why continuous better than discrete? Would love to see more reasoning beyond the empirical results.
3.	How does the training time compare across setups, especially when scaling up?
4.	In the inference time, how does the inference time step influence the model performance?
5.	Is the method scales well for image resolution? For example, finetune the model on a small, but large-resolution dataset. Current diffusion model often works well for much larger resolution like 512 and 1024. The 256^2 image generated in the paper is somehow limited for real application.
6.	A very important question for me is not confirming the scaling law of AR-based model for image generation(always, larger compute give better results). Rather, we are interested in comparing the diffusion model and the AR in term of the speed of scaling. This paper somehow demonstrates Fluid scales better than causal, discrete-token AR model. If it is possible, a comparing with pure diffusion model is very important.
7.	Are there any failure cases for the model?
8.	Not exactly a research problem, but will the code be open-sourced? Because this paper is mostly an empirical study, sharing the code and model would be the most important contribution than reading the paper. I personally think this is a key consideration when judging this paper.
One minor suggestion: While using a random scan order prevents the typical use of KV caching for causal sequences, an alternative could be to adopt a structured order, such as a grid-like pattern. This way, you could still cache info from nearby patches and keep some benefits of caching while generating the current patch.

---

> ### Author Response · Authors · 2024-11-19
> **Response to Reviewer UT5L (1/2)**
>
> Thank you for your insightful comments. Besides the general response, we have also incorporated your suggestions into the revised manuscript and provided detailed explanations for each question below. We hope that our revisions and rebuttal could address your concerns and that you might consider raising the rating. Please let us know if you have any further questions or require additional results
>
> ***Q2. Explanation of why continuous is better***
>
> We agree that more theoretical reasoning would strengthen our findings. Here, we provide additional quantitative and qualitative analysis to explain why continuous tokens and random-order generation outperform discrete tokens and raster-order generation:
>
> *Why continuous tokens perform better than discrete tokens*
>
> -   Compression Ratio Analysis: Continuous tokens preserve significantly more information than discrete tokens due to differences in compression ratios:
> 	 -  Continuous tokenizers map a 256x256 image to a 32×32×4 latent space (float32), yielding a compression ratio of approximately 2 bits per pixel.
> 	 - Discrete tokenizers map the same image to a 16×16 latent space with a codebook of size 8192 ($\log⁡_2(8192)=13$ bits per token), resulting in a much lower compression ratio of approximately 0.05 bits per pixel.
> 	 - This compression ratio for discrete tokens is even significantly lower than that of standard JPEG (typically 0.25–2 bits per pixel), making discrete tokens overly lossy and unable to retain sufficient image details.
> - Reconstruction Quality: As shown in Figure 4 of the main paper, reconstruction results using discrete tokens fail to preserve fine details, such as the identity and texture in the Mona Lisa example, leading to poorer visual fidelity and lower PSNR values. Continuous tokens, by contrast, provide much better reconstructions, retaining finer details and textures, which is beneficial for higher-quality image generation.
>
> *Gradient flow and expressive capacity*
>    - Continuous tokens benefit from smoother gradient flow due to their continuous-valued representation, enabling more precise optimization during training. Discrete tokens, on the other hand, rely on categorical representations, which can introduce challenges in optimization, such as difficulty in capturing nuanced details and gradients being limited by discrete choices.
>
> We believe this analysis complements the empirical findings presented in the paper and adds depth to our understanding of why continuous tokens and random-order generation yield better performance. Thank you for encouraging us to provide this additional reasoning.
>
>
> ***Impact of dataset size scaling***
>
> Thank you for highlighting this important aspect of scaling. To address this, we trained a 1.1B Fluid model with different amounts of image-text pairs for 500k iterations, and summarized the results in the table below.
>
> <Table S1. Scaling behavior of Fluid 1.1B model on different data sizes>
>
> | Dataset Portion (%) |   0.05 |    0.1 |   0.25 |    0.5 |      1 |      5 |     10 |    100 |
> |:---------------------|-------:|-------:|-------:|-------:|-------:|-------:|-------:|-------:|
> | FID                 |  52.70 |  12.33 |   7.72 |   7.23 |   6.74 |   6.70 |   6.67 |   6.73 |
>
> Our findings show that FID improves consistently as the dataset size increases, up to approximately 5-10% of the total training samples. Beyond this point, the improvements on FID plateau, likely because larger datasets require more training iterations and larger models to showcase its benefits.
>
> ***Q3: Training time comparison***
>
> We perform a comparison of the training FLOPs using different configurations. Models with continuous tokens require slightly more training FLOPs because of the DiffLoss MLP training:
>
> <Table S2. Training efficiency comparison on the 3.1B model across different setups.>
>
> | Token      | Order  | FLOPs    |
> |--------|----------|----------------|
> | Continuous | Random | 1.11E+16 |
> | Continuous | Raster | 1.11E+16 |
> | Discrete   | Random | 8.79E+15 |
> | Discrete   | Raster | 8.79E+15 |
>
> ***Q4: Inference step effect***
>
> For raster-order generation, inference steps are fixed since tokens must be generated one at a time with causal attention, resulting in 256 inference steps. For random-order generation, we conducted an analysis of FID versus inference steps for continuous tokens on the 3.1B model, as shown in the following table.
>
> <Table S3. FID vs inference steps on 3.1B model with continuous tokens>
>
> | Steps |     8 |   16 |   32 |   64 |  128 |  256 |
> |-------|------:|-----:|-----:|-----:|-----:|-----:|
> | FID   | 39.22 | 7.37 | 6.53 | 6.41 | 6.41 | 6.45 |
>
> The results indicate that FID consistently decreases with an increasing number of inference steps, up to 64 steps. Beyond 64 steps, further increases do not improve FID.

---

> ### Author Response · Authors · 2024-11-19
> **Response to Reviewer UT5L (2/2)**
>
> ***Q5: Finetune for higher resolution***
>
> Fluid models are indeed capable of generating 512x512 resolution images by fine-tuning the 256x256 checkpoints through position embedding interpolation. For 256x256 images, the token length is 256, whereas for 512x512 images, the token length increases to 1024.
> Due to the limited time during the discussion period, we focused on training a 3.1B Fluid model with continuous tokens and random-order generation at 512x512 resolution. We replace the trained position embedding with 2D rotary position embedding to support variable sequence length. Starting from the 256x256 checkpoint, we interpolated the positional embeddings to support the increased token length and fine-tuned the model for an additional 500k steps on 512x512 resolution images.
> As detailed in Section C.3 in the **updated supplementary material**, Figure A10 and A11, the 512x512 images demonstrate significant improvements in detail and sharpness compared to 256x256 images. This shows that Fluid models scale effectively with resolution, making them suitable for applications requiring higher-resolution image generation.
>
> ***Q6: Compare scaling with pure diffusion***
>
> The scaling of diffusion-based models, and later flow-based models, has been studied in several works in the literature. Most notably, in SD3 [1] (one of the current state-of-the-art flow-based models), they scaled the model from 0.8B to 8B parameters and observed consistent scaling behavior. In Figure A5 in the **updated supplementary material** and Table S4 below, we compare the scaling behavior of SD3 and Fluid in terms of the GenEval score. Surprisingly, despite various differences in model architecture, training data, and implementations, the scaling rates of these two models are quite similar within the same GenEval score range (0.6–0.7). Moreover, Fluid performs even better than SD3 at the same number of parameters. However, we also observe that the GenEval scores plateau for the 10.5B Fluid model compared to the 3.1B Fluid model. We hypothesize that this is due to the inconsistency between the scaling of the validation loss and the validation metrics. The scaling behavior of SD3 beyond 10B parameters would also be interesting to explore, and we leave it for future work.
>
>
> <Table S4. Scaling comparison between Fluid and SD3.>
>
> | SD3 Size (B) | SD3 GenEval | Fluid Size (B) | Fluid GenEval |
> |--------------|-------------|----------------|---------------|
> |         0.84 |        0.58 | 0.4B           |          0.62 |
> |            2 |        0.62 | 0.7B           |          0.65 |
> |            4 |        0.64 | 1.1B           |          0.67 |
> |            8 |        0.68 | 3.1B           |           0.7 |
>
> ***Q7: Failure cases***
>
> We have included examples of failure cases for both continuous tokens with random-order and raster-order generation Section D, Figure A15 and A16 in the **updated supplementary material**.
>
> - Random-Order Generation:
>     Occasionally, a bright spot appears in the generated image, rendering other parts unrecognizable. We hypothesize that this issue stems from insufficient diffusion steps in the diffusion head. Increasing the number of diffusion steps could potentially alleviate this problem.
> -  Raster-Order Generation:
>     In some cases, the generated images collapse into a gray pattern in the lower part, particularly at transitions between lines. We believe this occurs because the learned position embeddings struggle to capture the abrupt changes between lines. Incorporating 2D Rotary position embeddings may address this issue by better modeling spatial relationships.
>
> These observations provide valuable insights for future improvements and highlight areas for further exploration.
>
> ***Q8: Reproducibility***
>
> Thank you for highlighting the importance of code availability. We strictly adhere to the code implementation of MAR [2] (for continuous tokens) and MaskGIT [3] (for discrete tokens), and we have provided all additional hyperparameters and training details in the paper to ensure reproducibility. If there are specific details that you feel are missing or unclear, please let us know, and we would be happy to provide further clarifications or supplementary materials to support reproduction.
>
> [1] Scaling Rectified Flow Transformers for High-Resolution Image Synthesis, Esser et al., 2024
>
> [2] Autoregressive Image Generation without Vector Quantization, Li et al., 2024.
>
> [3] MaskGIT: Masked Generative Image Transformer, Chang et al., 2022

---

> ### Author Response · Authors · 2024-11-23
> **Look forward to your post-rebuttal feedback!**
>
> Dear Reviewer UT5L,
>
> Thanks again for your insightful suggestions and comments. Since the deadline of discussion is approaching, we are happy to provide any additional clarification that you may need.
>
> In our previous response, we carefully addressed your feedback and made the following updates and additions:
> Provided additional explanation of our empirical novelty and contributions to the community.
> - Offered detailed reasoning on why continuous tokens outperform discrete tokens.
> - Included a comparison of the scaling behavior of our Fluid model with diffusion models.
> - Conducted additional experiments to examine the effect of inference steps.
> - Conducted additional experiments to evaluate the performance of higher-resolution image generation (512x512).
> - Compared training efficiency across different configurations.
> - Discussed failure cases in detail.
>
> We hope these new experiments and explanations have clarified the contributions and strengthened our submission.
>
> Please do not hesitate to reach out if there are any further clarifications or analyses we can provide.
>
> Thank you for your time and thoughtful feedback!
>
> Best,
> Authors

---

> > ### Comment · Reviewer_UT5L · 2024-11-26
> > **Thanks for the feedbacks**
> >
> > Hi Authors
> >
> > Truly appreciate your detailed explanations and efforts. I remain positive about the evaluation of this work.
> >
> > Would love to hear the thoughts of other, though.
> >
> > Reviewer

---

> > > ### Author Response · Authors · 2024-11-27
> > > **Thank you!**
> > >
> > > We sincerely appreciate your positive feedback! We are happy to hear that your concerns are resolved.
> > >
> > > Thanks again for your insightful comments, please do not hesitate to let us know If there are any further clarifications or experiments we can offer.

---

### Author Response · Authors · 2024-11-19
**General Response: Contributions and New Experiments**

We sincerely thank all the reviewers for their insightful comments and constructive suggestions to strengthen our work. In addition to responding to specific reviewers, in this response, we would like to highlight our empirical novelty and contributions to the community, as well as the new experiments we have included in the rebuttal.

***Contributions***

Our paper does not aim to propose a novel technique but instead aims to address a critical and long-standing open question: whether autoregressive generative models are scalable in vision and how to achieve this scalability. Before our work, it was common wisdom that diffusion models are the de facto approach for visual generation. We challenge this notion and empirically demonstrate that autoregressive models are equally promising and can scale effectively. To the best of our knowledge, no prior work has systematically demonstrated this finding. Our extensive empirical results provide key design guidance and strong evidence to support autoregressive models in visual generation, opening up new opportunities for future research in this area. We list our key contributions and distinctions from prior works as follows:

1.  Empirical Validation of Scalability:
    Our primary contribution lies in empirically validating the scalability of autoregressive visual generative models under various design choices. Unlike prior works, which have lacked a systematic exploration of scaling properties, we provide a comprehensive study akin to the progression from GPT-1 and GPT-2 to GPT-3.

2.  Extensive Analysis of Key Design Choices:
   Previous works, such as MAR (Li et al., 2024), have explored scan order and token type comparisons only on small-scale models (e.g., 0.4 billion parameters) and on ImageNet datasets. In contrast, we systematically study the scaling behavior across these design choices—including discrete vs. continuous tokens and random vs. raster order—at significantly larger scales (up to 10.5 billion parameters). Our results reveal previously unknown insights: (1) all four variants of autoregressive models are scalable; (2) continuous tokens with random-order generation scale the best. We also provide extensive visualizations and analyses to thoroughly examine the pros and cons of different configurations. These insights offer valuable guidance for designing scalable autoregressive vision models in the future.

3.  Adapting Continuous Tokens for Text-to-Image Generation:
    We enhance the diffusion head architecture by incorporating cross-attention modules and text embedding layers to handle text-to-image generation tasks. This modification distinguishes our framework from MAR, which does not address text-conditioning in its design.

In a nutshell, our paper proposes the first autoregressive text-to-image generation system that scales up to 10.5 billion parameters. Our extensive experiments on two key design choices reveal that random order and continuous tokens play important roles in autoregressive visual generation. These insights from our paper are not only beneficial to text-to-image generation but also broadly impact the computer vision community. Our findings on scalability and scan order could shed light on the development of large vision-language multimodal models, where the autoregressive framework might be a more natural fit than diffusion models. Additionally, our findings on continuous tokens could benefit other visual generative tasks, such as video generation or 3D generation, where discrete tokenization is even more challenging and leads to greater information loss.

***Additional Experiments***

In this rebuttal, we have added more supporting experiments following reviewers’ suggestions. Here we summarize the new experiments that we added below:
-   Scaling with dataset size [UT5L, y1Ue]
-   Performance on higher resolution (512x512) [UT5L, hShT]
-   Comparison with LlamaGen [hShT]
-   Ablation studies on inference steps [UT5L]
-   Comparison of Inference time [QGyS, hShT]

We hope these new experiments can further complement and strengthen the empirical novelty and contributions of our work.

---

### Meta-Review · Area_Chair_nE24 · 2024-12-25

**Metareview:**

This paper investigates scaling behavior in autoregressive text-to-image models, focusing on token representation (discrete vs continuous) and generation order (random vs raster). The key findings show that continuous tokens with random-order generation scale most effectively, enabling state-of-the-art results with their 10.5B Fluid model. The paper demonstrates strong empirical results and thorough analysis, with reviews highlighting clear presentation and comprehensive experiments. While initial concerns were raised about novelty and resolution limitations, the authors provided strong responses during rebuttal: They demonstrated their model's effectiveness at 512x512 resolution achieving comparable FID and GenEval scores to 256x256, conducted new experiments showing benefits over recent work like LlamaGen, and clarified their key contribution of being first to systematically study scaling of autoregressive vision models at large scale (up to 10.5B parameters). The review scores (initially 5, 6, 6, 6) improved during discussion as reviewers were convinced by the additional experiments and clarifications. Though some concerns remain about theoretical justification, the thorough empirical validation and clear practical importance of the findings support accepting this paper, which offers valuable guidance for future large-scale autoregressive vision models. Thus I vote to accept this paper.

**Additional Comments On Reviewer Discussion:**

The reviewers raised several significant points that led to productive discussion. Reviewer UT5L requested explanation of why continuous tokens outperform discrete ones - the authors provided detailed analysis showing better compression ratios and gradient flow. Reviewer y1Ue asked for scaling law analysis - the authors demonstrated consistent improvements with dataset size up to 5-10% of total data. Reviewer QGyS questioned fairness of comparisons - the authors detailed careful controls across configurations and provided inference speed benchmarks. Reviewer hShT raised concerns about resolution - the authors added 512x512 experiments showing comparable performance to 256x256. The discussion was particularly active around resolution comparisons and practical significance, with hShT initially disagreeing about fairness across resolutions but ultimately being convinced by new 512x512 results showing strong performance. Multiple reviewers explicitly acknowledged satisfaction with the responses, with y1Ue indicating intent to raise their score (though not yet reflected) and hShT increasing their score. The thorough engagement with reviewer feedback through new experiments and analyses strengthened the paper's empirical validation significantly. Also reviewer y1Ue said he would raise his scores but didn't.

---

### Decision · Program_Chairs · 2025-01-22

Accept (Poster)